# Yeast translation elongation factor eEF3 promotes late stages of tRNA translocation

Namit Ranjan[1,*,†] , Agnieszka A Pochopien[2,3,†], Colin Chih-Chien Wu[4,†], Bertrand Beckert[3], Sandra Blanchet[1], Rachel Green[4,5,**] , Marina V Rodnina[1,***] & Daniel N Wilson[2,3,****]

## Abstract

In addition to the conserved translation elongation factors eEF1A and eEF2, fungi require a third essential elongation factor, eEF3. While eEF3 has been implicated in tRNA binding and release at the ribosomal A and E sites, its exact mechanism of action is unclear. Here, we show that eEF3 acts at the mRNA–tRNA translocation step by promoting the dissociation of the tRNA from the E site, but independent of aminoacyl-tRNA recruitment to the A site. Depletion of eEF3 *in vivo* leads to a general slowdown in translation elongation due to accumulation of ribosomes with an occupied A site. Cryo-EM analysis of native eEF3-ribosome complexes shows that eEF3 facilitates late steps of translocation by favoring non-rotated ribosomal states, as well as by opening the L1 stalk to release the E-site tRNA. Additionally, our analysis provides structural insights into novel translation elongation states, enabling presentation of a revised yeast translation elongation cycle.

**Keywords** ABC ATPase; cryo-EM; eEF3; E-site tRNA; L1 stalk
**Subject Category** Translation & Protein Quality
**The EMBO Journal (2021) 40: e106449**

## Introduction

Protein synthesis is an evolutionary conserved process. Translation elongation entails the same repetitive steps of decoding, peptide bond formation, and translocation in all living organisms, catalyzed by highly homologous translation elongation factors eEF1A/EF-Tu and eEF2/EF-G in eukaryotes and bacteria, respectively, assisted by eEF1B/EF-Ts and eIF5A/EF-P (Dever & Green, 2012; Rodnina, 2018). It is therefore particularly surprising that some fungi, such as

*Saccharomyces cerevisiae, Candida albicans,* and *Pneumocystis carinii* have an additional essential, abundant elongation factor, eEF3 (Skogerson & Wakatama, 1976; Uritani & Miyazaki, 1988; Kamath & Chakraburtty, 1989; Colthurst *et al*, 1991). Moreover, recent bioinformatic analyses indicate that eEF3-like proteins exist more broadly in unicellular eukaryotes, not just in fungi (Mateyak *et al*, 2018). Some organisms, e.g., *S. cerevisiae*, also contains an eEF3 homologue, termed New1p, which when overexpressed can rescue strains depleted for eEF3 (Kasari *et al*, 2019b). In exponentially growing *S. cerevisiae* cells, eEF3, eEF1A, eEF2, and eRF1 exert the strongest control over the rate of translation (Firczuk *et al*, 2013). In yeast, the amount of eEF3 per cell is comparable to that of ribosomes, eEF2, and eEF1B, 10 times lower than that of eEF1A, and 5-10 times higher than of the termination factors and the majority of initiation factors (Firczuk *et al*, 2013). These data suggest that there is sufficient eEF3 to participate in every round of translation elongation.

eEF3 is a cytoplasmic protein that belongs to the superfamily of ATP-binding cassette (ABC) proteins, specifically, of the F subtype (Murina *et al*, 2018). In *S. cerevisiae*, eEF3 consists of a single polypeptide chain of 1,044 amino acids comprising several domains: a HEAT repeat domain, a four-helix bundle domain (4HB domain), two ATP-binding domains, ABC1 and ABC2, with a chromodomain (CD) insertion within ABC2, as well as eEF3-specific C-terminal domain (Kambampati *et al*, 2000; Murina *et al*, 2018). A low-resolution (9.9 Å) cryo-electron microscopy (cryo-EM) structure of the ribosome–eEF3 complex revealed that eEF3 binds to the ribosome, spanning across the boundary between the head of the 40S subunit and the central protuberance of the 60S subunit, where it contacts ribosomal RNA (rRNA) and proteins, but should not interfere with the binding of eEF1A or eEF2 (Andersen *et al*, 2006). Despite these structural clues, the function of eEF3 remains poorly understood. Early studies reported that eEF3 facilitates binding of the ternary complex eEF1A–GTP–aminoacyl-tRNA to the decoding site (A site) of the ribosome, thus increasing the fidelity of translation (Uritani & Miyazaki, 1988); genetic and physical interactions between eEF1A

1 Department of Physical Biochemistry, Max Planck Institute for Biophysical Chemistry, Göttingen, Germany
2 Gene Center, Department for Biochemistry and Center for integrated Protein Science Munich (CiPSM), University of Munich, Munich, Germany
3 Institute for Biochemistry and Molecular Biology, University of Hamburg, Hamburg, Germany
4 Department of Molecular Biology and Genetics, Johns Hopkins University School of Medicine, Baltimore, MD, USA
5 Howard Hughes Medical Institute, Johns Hopkins University School of Medicine, Baltimore, MD, USA
*Corresponding author. Tel: +49 551 2012958; E-mail: namit.ranjan@mpibpc.mpg.de
**Corresponding author. Tel: +1 410 614 4928; E-mail: ragreen@jhmi.edu
***Corresponding author. Tel: +49 551 2012900; E-mail: rodnina@mpibpc.mpg.de
****Corresponding author. Tel: +49 40 42838 2847; E-mail: daniel.wilson@chemie.uni-hamburg.de
†These authors contributed equally to this work

and eEF3 appear to support this hypothesis (Anand *et al*, 2003; Anand *et al*, 2006). An *in vitro* study of tRNA binding to the ribosome argued that eEF3 accelerates dissociation of tRNA from the exit site (E site) independent of translocation and is required for aminoacyl-tRNA binding to the A site when the E site is occupied (Triana-Alonso *et al*, 1995); thus, according to this model, eEF3 regulates decoding. By contrast, another study proposed that eEF3 facilitates the release of tRNA from the P site during the disassembly of post-termination complexes at the ribosome recycling step of translation (Kurata *et al*, 2013).

eEF3 has also been implicated in translational control in response to environmental stress. Expression of eEF3 is altered during morphogenesis in *Aspergillus fumigatus* and upon starvation in *S. cerevisiae* (Grousl *et al*, 2013; Kubitschek-Barreira *et al*, 2013). Interaction of eEF3 with the ribosome in *S. cerevisiae* is affected by Stm1, a protein that binds to the mRNA entry tunnel of the ribosome (Ben-Shem *et al*, 2011), inhibits translation at the onset of the elongation cycle, and promotes decapping of a subclass of mRNAs (Van Dyke *et al*, 2009; Balagopal & Parker, 2011). In yeast lacking Stm1, eEF3 binding to ribosomes is enhanced and overexpression of eEF3 impairs growth (Van Dyke *et al*, 2009). On the other hand, eEF3 affects the function of Gcn1, thereby modulating the kinase activity of Gcn2 (Sattlegger & Hinnebusch, 2005). In addition to these functional interactions, the Saccharomyces genome and the STRING databases suggest that eEF3 interacts with a number of proteins that may affect the efficiency of translation (Cherry *et al*, 2012; Szklarczyk *et al*, 2019), including eEF2, heat shock proteins, or She2 and She3, the adaptor proteins that facilitate targeted transport of distinct mRNAs (Heym & Niessing, 2012). Together these data raise the possibility that eEF3 may act as a hub for translational control, but insight into the role that eEF3 plays within these regulatory networks is hampered by the lack of understanding of its basic function.

Here, we address the function of eEF3 using a combination of rapid kinetics in a fully reconstituted yeast *in vitro* translation system, ribosome profiling, and cryo-EM. First, we show using biochemical and kinetic approaches that eEF3 helps eEF2 to complete tRNA–mRNA translocation by accelerating the late stages of the process. While eEF3 facilitates E-site tRNA release, this activity is not dependent on the binding of ternary complex to the A site, as previously reported (Triana-Alonso *et al*, 1995). Ribosome profiling of strains conditionally depleted of eEF3 reveals global changes in ribosome state in the cell wherein ribosomes accumulate with an occupied A site (typical of ribosomes in a pre-translocation state). These data are consistent with the presence of a new rate-limiting step in elongation where ribosomes cannot undergo translocation due to the presence of deacylated tRNA in the E site. Cryo-EM of native eEF3-ribosome complexes provides a structural basis for the mechanism of action of eEF3 to promote late steps of translocation as well as facilitate E-site tRNA release.

## Results

### eEF3 promotes late steps of tRNA–mRNA translocation

To understand the function of eEF3, we used a fully reconstituted translation system from yeast components (Appendix Fig S1A). We assembled an 80S initiation complex (80S IC) using purified 40S and 60S subunits, mRNA, and initiator tRNA ([$^3$H]Met-tRNA$_i^{Met}$) in the presence of initiation factors eIF1, eIF1A, eIF2, eIF3, eIF5, and eIF5B (Appendix Fig S1B). Addition of [$^{14}$C]Phe-tRNA$^{Phe}$ in the ternary complex with eEF1A and GTP (TC-Phe), and eIF5A resulted in decoding of the next codon UUU and formation of the dipeptide MetPhe (Fig 1A). Using a fully reconstituted translation system allows us to study each step of translation in a codon-resolved manner and to identify the reactions that are promoted by eEF3. To test whether eEF3 affects the first round of translation elongation, we monitored the dipeptide formation in real time. As the reaction is rapid, we carried out experiments in a quench-flow apparatus. We separated dipeptides from unreacted amino acids by HPLC and quantified the extent of dipeptide formation by radioactivity counting. The rate of MetPhe formation (0.15 s$^{-1}$) was not affected by the addition of eEF3 (Fig 1B), and only slightly at very high eEF3 concentration (Fig EV1A). Similarly, formation of the MetVal peptide measured on an mRNA with Val (GUU) codon was not affected by the presence of eEF3 (Fig EV1B). These results indicate that eEF3 is not essential for the first round of decoding or peptide bond formation. Next, we tested whether the second round of elongation is affected. We prepared an 80S complex carrying a dipeptidyl-tRNA (MetPhe-tRNA$^{Phe}$) in the A site and a deacylated tRNA$_i^{Met}$ in the P site (denoted as 80S 2C for dipeptide complex), added a TC with Val-tRNA$^{Val}$ reading the GUU codon (TC-Val) and then monitored tripeptide formation (Fig 1A). In this case, the tripeptide MetPheVal was formed only when both eEF2 and eEF3 were added together with TC-Val (Fig 1C). The effect was less dramatic when Phe and Val codons were reverted, as some MetValPhe was also formed in the absence of eEF3, but the addition of eEF3 increased the rate of tripeptide formation by 10-fold, from 0.03/s to 0.3/s (Fig EV1C). Thus, eEF3 is required for the second—and presumably the following—rounds of translation elongation.

To form tripeptide, the ribosome has to move along the mRNA in a step referred to as translocation and then bind the next aminoacyl-tRNA (Fig 1A). In bacteria and higher eukaryotes, the factor that promotes translation, EF-G/eEF2, alone drives the reaction to completion. We then asked the question whether yeast eEF2 or eEF3 alone can promote translocation with longer incubation times. To test this, we incubated 80S 2C with eEF2 or eEF3 alone, or with both factors together, for 15 min and then added TC-Val (Fig 1D). The rate of tripeptide formation was very similar and high with eEF2, or eEF2 and eEF3, about 0.4/s, indicating that eEF2 alone can complete the translocation process given enough time, but to function within physiological timeframes, the reaction requires eEF3. This also indicates that eEF3 is not required for the decoding of the Val codon or peptide bond formation *per se*. Interestingly, eEF3 alone facilitated translocation, albeit only on a fraction of ribosomes, as seen by the endpoint defect (Fig 1D). We speculate that eEF3 on its own promoted tRNA translocation, albeit not to the full extent.

We next specifically studied the role of eEF3 in translocation. Translocation is a dynamic process that entails several steps. Analogous to the well-studied bacterial translocation process (Zhou *et al*, 2014; Belardinelli *et al*, 2016), translocation in yeast likely encompasses the following major steps. (1) Prior to eEF2 binding, the tRNAs move into hybrid states upon peptidyl transfer, with concomitant rotation of the ribosomal subunits. (2) eEF2 binds and

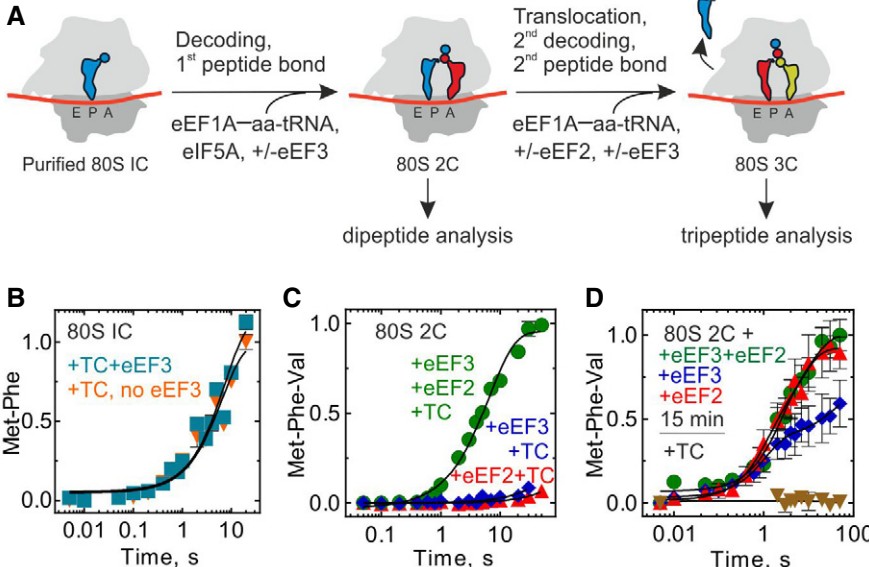

**Figure 1. eEF3 is essential for tri- and polypeptide formation.**

A Schematic of the translation elongation cycle as studied here.

B Met-Phe formation monitored upon rapidly mixing initiation complexes (80S IC) with ternary complexes eEF1A–GTP–[14C]Phe-tRNA$^{Phe}$ in the absence (orange, $0.13 \pm 0.02/s$) or presence (cyan, $0.15 \pm 0.02/s$) of eEF3 (2 μM) in a quench-flow apparatus, and the extent of peptide formation was analyzed by HPLC and radioactivity counting. Data are normalized to Met-Phe formation in the absence of eEF3 with the maximum value in the dataset set to 1. Data are presented as mean $\pm$ SEM of $n = 3$ biological replicates.

C Met-Phe-Val formation monitored upon rapidly mixing 80S complexes carrying MetPhe-tRNA$^{Phe}$ (80S 2C) with ternary complexes eEF1A–GTP–[14C]Val-tRNA$^{Val}$ in the presence of eEF2 and eEF3 (green, $0.15 \pm 0.01/s$), eEF2 (red), or eEF3 (blue). Data presented as mean $\pm$ SEM of $n = 3$ biological replicates.

D Met-Phe-Val formation monitored with 80S 2C incubated with eEF2 and eEF3 (green, $0.4 \pm 0.14/s$), eEF2 (red, $1.0 \pm 0.5/s$), or eEF3 (blue), or in the absence of eEF2 and eEF3 (brown) for 15 min before adding with ternary complexes eEF1A–GTP–[14C]Val-tRNA$^{Val}$ and monitoring the kinetics of Val incorporation. Data presented as mean $\pm$ SEM of $n = 3$.

Data information: Datasets in (C) and (D) are normalized to Met-Phe-Val formation in presence of eEF2 and eEF3 with the maximum value set to 1.
Source data are available online for this figure.

catalyzes movement of tRNAs into the so-called chimeric states, with displacement of the mRNA together with tRNA anticodons on the small subunit and accommodation of the CCA-3′ end of the peptidyl-tRNA in the P site (Fig 2A). (3) mRNA and the two tRNAs move to the E and P sites of the small subunit, the small subunit rotates backward relative to the large subunit, and eEF2 and the E-site tRNA dissociate from the ribosome (Fig 2A). One diagnostic test for the completion of step (2) is the reaction of peptidyl-tRNA with the antibiotic puromycin (Pmn). In the absence of translation (the 80S 2C complex), peptidyl-tRNA does not react with Pmn, even with very long incubation times (Fig 2B, brown triangles). Upon addition of eEF2 and eEF3, or eEF2 alone, peptidyl-tRNA reacts with Pmn. The rate of reaction with puromycin is indistinguishable from that observed for the post-translation complex prepared by prolonged pre-incubation with the factors (Fig EV1D–F), indicating that eEF2 alone is sufficient to facilitate the early stages of translocation through step (2). We note that in the presence of eEF3 and eEF2, the reaction with Pmn was faster than with eEF2 alone, and the endpoint of the reaction was somewhat lower, suggesting that binding of eEF3 to the ribosome stabilizes a ribosome conformation that influences the efficiency of Pmn reaction, possibly by affecting Pmn binding or by stabilizing peptidyl-tRNA in the exit tunnel (see cryo-EM data below). Also, consistent with the results of the tripeptide synthesis (Fig 1D), eEF3 alone appears to catalyze translocation,

but not to the same extent as in the presence of eEF2 (Fig 2B), which is not increased further with the pre-incubation time (not shown).

## E-site tRNA release by eEF3 does not require ternary complex binding

The final step (3) of translocation involves dissociation of deacylated tRNA from the E site (Fig 2A) (Belardinelli *et al*, 2016). To monitor this step in real time, we used a rapid kinetics assay that follows a fluorescence change of tRNA$^{fMet}$ labeled by fluorescein, tRNA$^{fMet}$(Flu), upon dissociation from the ribosome (Belardinelli *et al*, 2016). The tRNA is functionally active in the yeast translation system to the same extent as a non-modified tRNA$_i^{Met}$, as validated by the tripeptide formation assay (Fig EV1G). When we initiate translation by rapidly mixing 80S 2C containing tRNA$^{fMet}$(Flu) and MetPhe-tRNA$^{Phe}$ in the P and A sites, respectively, with eEF2, eEF3, and TC-Val in the stopped-flow apparatus, fluorescence decreases (Fig 2C, green trace). In the process of reaction, addition of eEF2 and eEF3 together with TC-Val leads to a first translocation, Val-tRNA$^{Val}$ binding, next peptide bond formation, followed by a second translocation and the dissociation of tRNA$^{Met}$ from the E site. At the starting point of the experiment, tRNA$^{fMet}$(Flu) resides in the P/P or P/E state. The fluorescence change over the time course with eEF2,

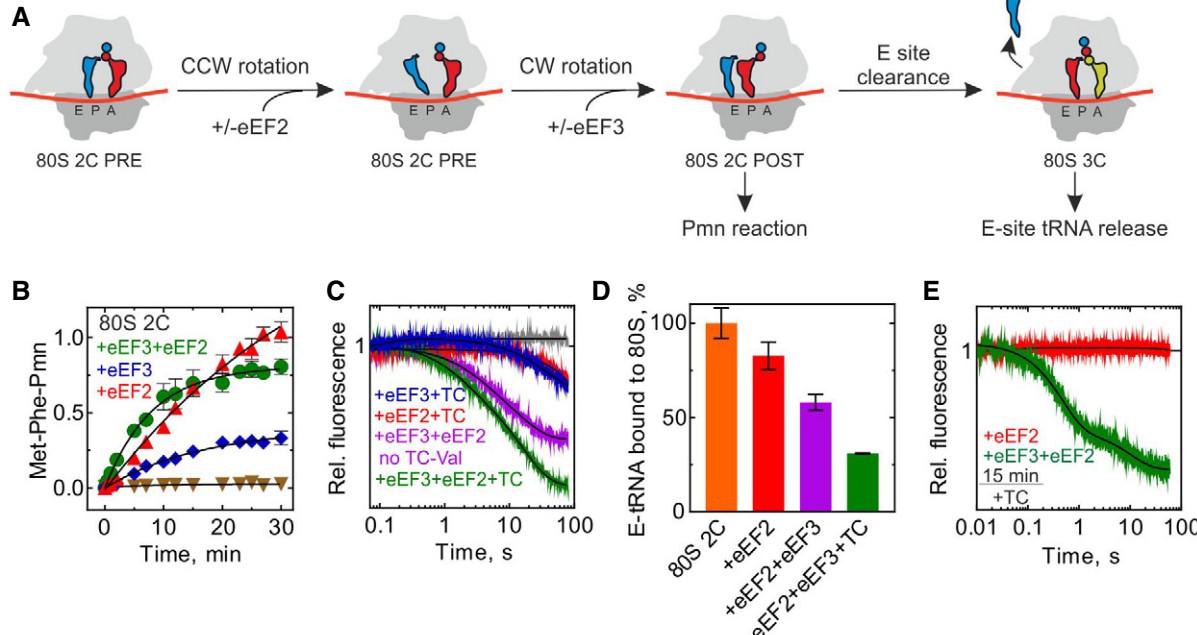

**Figure 2. eEF3 in tRNA movement during translocation and E-site clearance.**

A  Schematic of tRNA translocation.

B  mRNA–tRNA translocation monitored in a time-resolved Pmn assay upon mixing 80S 2C with Pmn in the absence (brown triangles) or presence of eEF2 and eEF3 (green, 0.13 ± 0.01/min), eEF2 (red, 0.030 ± 0.005/min), or eEF3 (blue, 0.10 ± 0.01/min). Data are normalized to Met-Phe-Pmn formation in presence of eEF2 with the maximum value set to 1. Data presented as mean ± SEM of $n = 3$.

C  Dissociation of deacylated tRNA from the E site by the fluorescence change of tRNA$^{fMet}$(Flu) upon rapidly mixing of 80S 2C with buffer (gray), or with ternary complexes eEF1A–GTP–[$^{14}$C]Val-tRNA$^{Val}$ in the presence of eEF2 and eEF3 (green), eEF2 (red), eEF3 (blue), or eEF2 and eEF3 without the ternary complex (magenta) in a stopped-flow apparatus. Each trace is an average of 5–7 individual time courses and normalized at 1 for the fluorescence at the start of the reaction.

D  tRNA$^{fMet}$(Flu) co-eluting with 80S 2C incubated without factors (orange), with eEF2 (red), with eEF2 and eEF3 (magenta), or with eEF2, eEF3, and TC-Val (green) for 15 min before loading on a BioSuite 450 size-exclusion column to separate ribosome-bound and ribosome-unbound tRNA. Data are normalized to tRNA$^{fMet}$(Flu) bound to 80S 2C in the absence of eEF2/eEF3 with the maximum value set to 100%. Data presented as mean ± SEM of $n = 3$.

E  Dissociation of tRNA$^{fMet}$(Flu) from 80S 2C deacylated tRNA incubated with eEF2 and eEF3 (green) or eEF2 (red) for 15 min before adding with ternary complexes. Each trace is an average of 5–7 individual time courses and normalized at 1 for the fluorescence at the start of the reaction.

Source data are available online for this figure.

eEF3, and TC-Val gives the maximum signal difference anticipated upon E-site tRNA dissociation (Fig 2C, green trace). When a similar experiment is carried out in the absence of TC-Val, only the first translocation can occur; we observe that in this case, the fluorescence of tRNA$^{fMet}$ decreases considerably, but not to the same extent as when TC-Val is included (Fig 2C, magenta trace). This can reflect movement of tRNA$^{fMet}$(Flu) to the E site, where it has a somewhat lower fluorescence than in the P site, or a movement to the E site followed by partial dissociation of the tRNA from the ribosome. Addition of eEF3 or eEF2 alone resulted in only a very small fluorescence change (Fig 2C, blue and red traces). These results suggest that eEF2 and eEF3 together facilitate the movement of the E-site tRNA and its ultimate dissociation from the E site. By contrast, eEF2 alone can catalyze partial tRNA displacement (that promotes Pmn reactivity, Fig 2B), but is incapable of promoting dissociation of the E-site tRNA during the short (physiologically relevant) time window.

We further tested whether eEF2 is capable of releasing deacylated tRNA over a long incubation time, as in the tripeptide formation experiment. We incubated 80S 2C (with tRNA$^{fMet}$(Flu) and

MetPhe-tRNA$^{Phe}$) with eEF2 and eEF3 or with eEF2 alone for an extended period (15 min) and monitored how much tRNA$^{fMet}$(Flu) is retained by the ribosome by following the fluorescence co-eluting with the ribosomes during gel filtration (Fig 2D). Upon 15-min incubation with eEF2 alone, very little tRNA is released (Fig 2D, orange and red bars). Upon incubation with eEF2 and eEF3, a significant fraction of tRNA$^{fMet}$(Flu) is released during gel filtration, suggesting labile binding of the E-site tRNA (magenta). Addition of eEF2, eEF3, and TC-Val, which results in two rounds of translocation and should remove the E-site tRNA from all active ribosomes, results in about 70% tRNA$^{fMet}$(Flu) dissociation (Fig 2D, green bar). To further test the E-site dissociation in a time-resolved way, we also monitored tRNA$^{fMet}$(Flu) dissociation in a stopped-flow experiment (Fig 2E). No fluorescence change is observed when TC-Val is added to 80S 2C incubated with eEF2 alone (Fig 2E, red trace), although from the kinetics of tripeptide formation we observe that under these conditions, eEF2 alone promotes partial tRNA translocation (Fig 1D, red triangles). These experiments clearly establish that binding of the cognate TC into the A site following eEF-2 catalyzed translocation is not sufficient to promote dissociation of the E-site tRNA, in contrast

to previous suggestions (Triana-Alonso *et al*, 1995). Thus, eEF2 promotes partial translocation by facilitating movement of peptidyl-tRNA from the A to P site, which also implies that the deacylated tRNA has to move toward the E site, but does not allow the release of the deacylated tRNA from the ribosome. The role of eEF3 is to facilitate the late stages of translocation by promoting the dissociation of the deacylated tRNA from the E site, but has little effect on A-site binding (Figs 1D and 2D).

## Ribosome profiling of eEF3-depleted yeast cells

Having shown *in vitro* that eEF3 is essential for late steps of translocation, we corroborated our findings *in vivo* using polysome and ribosome profiling. Because eEF3 is an essential gene in budding yeast, to observe its function *in vivo* we developed a system to conditionally deplete eEF3 using transcriptional shutoff and an auxin degron tag (mAID) as described previously (Nishimura *et al*, 2009; Schuller *et al*, 2017). In the eEF3 depletion strain (dubbed eEF3d), expression of eEF3 is controlled by a *GAL1* promoter that is turned off in the presence of glucose. Following a switch to glucose in the media and the addition of auxin, eEF3 was not detectable by immunoblotting after 8 h (Fig EV2A). As anticipated based on its essential nature, depletion of eEF3 reduces cell growth compared to wild-type (WT) cells (Fig EV2B). Furthermore, we performed polysome run-off experiments (i.e., without elongation inhibitors included during lysis) and observed slightly increased polysome-to-monosome ratios, indicative of defective translation elongation in the absence of eEF3 (Fig EV2C).

We next prepared libraries of ribosome footprints from WT and eEF3d strains after 8 h of conditional growth. Our recent study showed that using combinations of elongation inhibitors for library preparations allows us to distinguish three distinct ribosome functional states—pre-accommodation (PreAcc), pre-peptide bond formation (PrePT), and pre-translocation (PreTrans)—from different sized ribosome-protected footprints (RPFs) (Wu *et al*, 2019b) (Fig 3A). In a first set of experiments, we prepared libraries with a combination of cycloheximide (CHX) and tigecycline (TIG) added to cellular lysates to prevent interconversion between ribosome elongation states; CHX blocks translocation whereas TIG blocks aa-tRNA accommodation in the A site as indicated (Schneider-Poetsch *et al*, 2010; Jenner *et al*, 2013) (Fig 3A). In these libraries, 21 nt RPFs correspond solely to ribosomes with an empty A site on the small subunit, e.g., in a PreAcc state, whereas 28 nt RPFs predominantly correspond to ribosomes trapped in a PreTrans state (Fig 3A). In WT cells, we observe a distribution of 21 and 28 nt RPFs genome-wide, suggesting that no single step of translation elongation is rate-limiting under normal circumstances, but that instead ribosomes are found in a relatively even distribution of functional states (Fig 3B, black line). By contrast, in eEF3d cells, we observed a drastic reduction in 21 nt RPFs (and a concomitant increase in 28 nt RPFs) when compared to WT cells (Fig 3B), indicating an accumulation of ribosomes in either a PrePT or PreTrans state. This observation establishes a role for eEF3 in translation elongation, as its depletion causes a genome-wide change in the distribution of ribosome functional states. More specifically, the reduction in 21 nt RPFs in eEF3d cells is indicative of an increase in ribosomes with A sites occupied by aa-tRNAs. These data suggest that either peptide bond formation or translocation becomes rate-limiting *in vivo* in the absence of eEF3.

To determine the *in vivo* rate-limiting step (i.e., peptide bond formation or translocation) in the absence of eEF3, we next prepared libraries using a combination of CHX and anisomycin (ANS). Because ANS blocks peptidyl transfer (Grollman, 1967), 28 nt RPFs in these libraries report solely on ribosomes in a PreTrans state, whereas 21 nt RPFs correspond to ribosomes either in a PreAcc or PrePT state due to dissociation of the unreacted aa-tRNA from the A site (Fig 3C). As anticipated, libraries prepared from WT cells revealed a distribution of 21 and 28 nt RPFs, but with a slightly greater number of 21 nt RPFs than libraries prepared with CHX + TIG (Wu *et al*, 2019b). We further found that the footprints from eEF3d cells are almost devoid of 21 nt RPFs when compared to the WT samples prepared with CHX + ANS (Fig 3D). These data indicate that neither tRNA selection nor peptide bond formation is rate limiting in the absence of eEF3 but rather a step specific to translocation. In addition, we found that the modest association between eEF1A or eEF2 and the elongating ribosomes was not influenced by eEF3 depletion as monitored by polysome profiles and immunoblotting (Fig EV2D). Taken together, our experiments argue that eEF3 promotes the translocation step of elongation, consistent with our *in vitro* data (Fig 2B).

## Codon level analysis of ribosome functional states in the eEF3d strain

In addition to global changes in the distribution of ribosome functional states, we asked whether ribosomes are disproportionally stalled at specific sites. By calculating A-site ribosome occupancies (i.e., pause scores) of 21 nt RPFs (from samples prepared with CHX + TIG) at all 61 sense codons, we observed a distribution of pause scores in WT cells, and a drastic reduction in this distribution in eEF3d cells; as a result, we see a nearly horizontal line comparing these different distributions with a slope of 0.18 (Fig 3E). Different ribosome occupancies on codons, i.e., "codon optimality", is thought to be determined by several factors including codon usage, cellular tRNA concentrations, and the efficiency of wobble pairing (Percudani *et al*, 1997; dos Reis *et al*, 2004). For example, pause scores at rare codons (such as CGA) are substantially higher than those at common codons (such as AGG) in WT cell (Fig 3E and F), as shown in earlier studies (Weinberg *et al*, 2016; Wu *et al*, 2019b); these differences are however substantially smaller in eEF3d cells (Fig 3F, quantified by variance). Importantly, because the calculation of ribosome occupancies (pause scores) at codons from ribosome profiling experiments relies on internal normalization (i.e., an increase in ribosome occupancy at one codon necessarily reduces those at other codon(s)), we do not interpret the absolute values of these numbers in any manner. Given that 21 nt RPFs from samples prepared with CHX + TIG represent ribosomes poised for decoding (Fig 3A), these observations suggest that in the absence of eEF3, a step preceding decoding has become rate limiting, thereby reducing the decoding rate in the A site in a global manner such that codon-specific differences are lost (Fig 3F, e.g., AGG versus CGA).

We similarly characterized 28 nt RPFs in libraries prepared with CHX + ANS (Fig 3D), focusing on whether there are any sequence motifs that tend to be enriched in the PreTrans state (Fig 3C); in these libraries, 28 nt RPFs represent ribosomes trapped in a

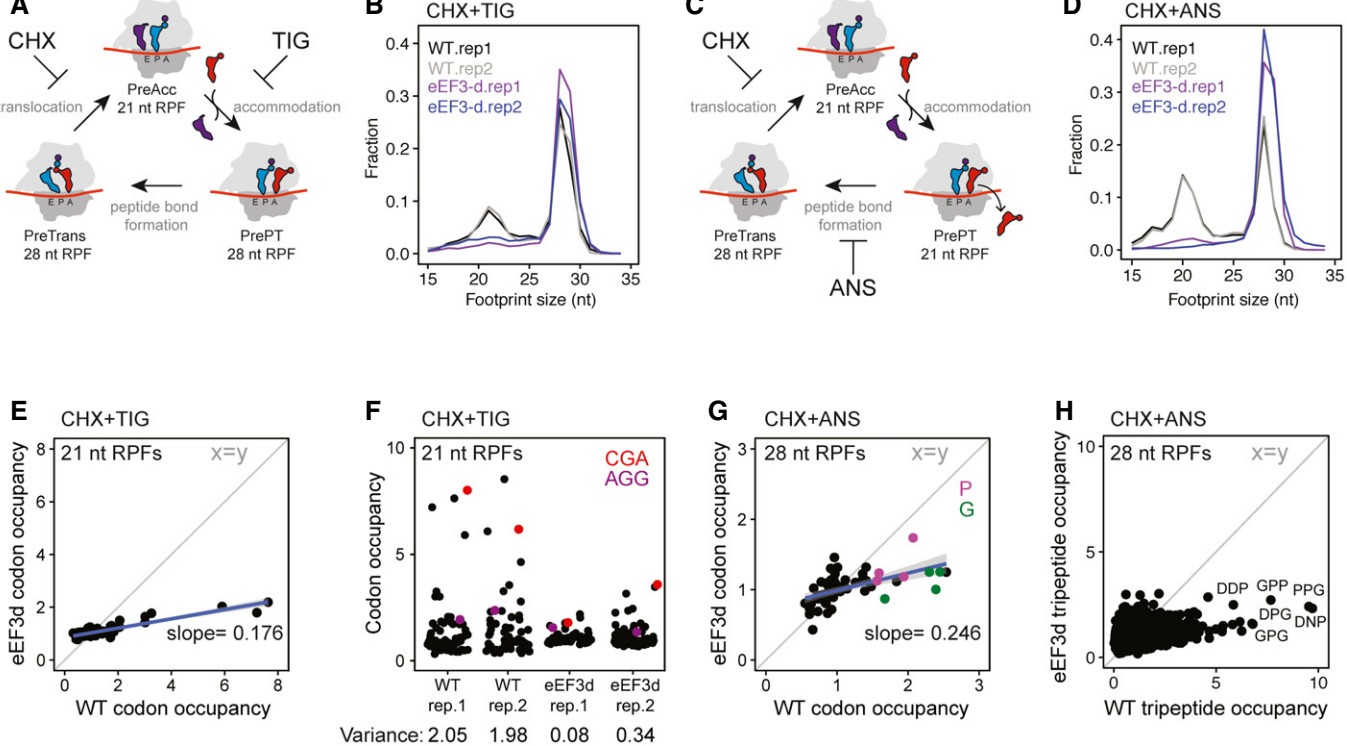

**Figure 3. eEF3 depletion perturbs translation elongation transcriptome-wide.**

A   Schematic representation of the eukaryotic elongation cycle with ribosome footprint sizes to illustrate the functional states isolated by CHX + TIG. PreAcc, pre-accommodation; PrePT, pre-peptide bond formation; PreTrans, pre-translation.

B   Size distributions of ribosome footprints for WT (black and gray) and eEF3d (purple and blue) cells from libraries prepared with CHX + TIG. Two biological replicates are shown.

C   Similar to (A), with footprint sizes isolated by CHX + ANS.

D   Similar to (B), from libraries prepared with CHX + ANS.

E   Scatter plot of codon-specific ribosome occupancies for 21 nt RPFs comparing eEF3d to WT cells from libraries prepared with CHX + TIG. Trend line (blue) and its slope are shown. The diagonal line indicates the distribution expected for no change.

F   Codon-specific ribosome occupancies for 21 nt RPFs from WT and eEF3d cells (two biological replicates). CGA and AGG codons are shown in red and purple, respectively. Variance for each dataset is indicated.

G   Scatter plot of codon-specific occupancies for 28 nt RPFs comparing eEF3d to WT cells from libraries prepared with CHX + ANS. Trend line (blue) and its slope are shown. Codons that encode proline and glycine are colored in purple and green, respectively.

H   28 nt RPF-ribosome occupancies of 5,771 tripeptide motifs are plotted for WT and eEF3d cells, for motifs more than 100 occurrences in yeast transcriptome. Motif sequences enriched upon eIF5A depletion are labeled.

PreTrans state. First, in the WT cells, we found a distribution of codon-specific occupancies that differ by as much as threefold; among the most enriched codons are those that correspond to those encoding proline (P, purple) and glycine (G, green) (Fig 3G). Second, as for our analysis of the 21 nt RPFs (Fig 3E), we observed a loss of codon-specific differences in ribosome occupancy when eEF3 is depleted, as evidenced by a flattening of the correlation plot (slope 0.25). Further analysis of peptide motifs revealed that among 8,000 tripeptides, the most enriched sequences were those encoding collections of P, G, and D amino acids (Figs 3H and EV2E). These observations are reminiscent of sequences that depend on eIF5A for peptide bond formation (Pelechano & Alepuz, 2017), (Schuller *et al*, 2017), though here we are seeing that these same sequences are slowing the translocation step of protein synthesis. Consistent with the results of *in vitro* biochemistry, we conclude that eEF3 contributes to the translocation step of protein synthesis, and in its absence, an early step in the process becomes rate limiting.

**A native structure of eEF3 on the yeast 80S ribosome**

To investigate a physiologically relevant structure of eEF3 on the yeast 80S ribosome at different phases of the elongation cycle, we opted to obtain eEF3-80S complexes by employing a native pull-down approach using C-terminally TAP-tagged eEF3. The distant location of the C terminus with respect to the ribosome (Appendix Fig S2A and B), and the lack of effect of the C-terminal TAP-tag has on growth compared to the wild-type strain (Appendix Fig S2C–E), suggests that it has little if any effect on eEF3 function and therefore would be appropriate for obtained native eEF3-80S complexes. Thus, to isolate native eEF3-80S complexes, a *S. cerevisae* BY4741 strain expressing C-terminally TAP-tagged eEF3 was grown to log phase and eEF3-80S complexes were isolated using IgG-coupled magnetic beads (Fig 4A). The complex was eluted by TEV protease cleavage of the TAP-tag in the presence or absence of non-hydrolyzable ADPNP, and, in both cases SDS–PAGE

analysis indicated the presence of ribosomal proteins in addition to eEF3 (Appendix Fig S2F). By contrast, no ribosomal proteins were observed in a control purification performed using an untagged wild-type strain, suggesting that the ribosomal complexes observed in the TAP-tagged strains co-purified with the eEF3 (Appendix Fig S2F). The eluate containing the native eEF3-80S complexes, in the presence of ADPNP, was cross-linked with glutaraldehyde and directly applied to cryo-grids. It is important to note that in the absence of cross-linker, reconstructions yielded 80S ribosomes, but without additional density for eEF3, suggesting that eEF3 dissociated from the ribosomes during grid application. The native eEF3-80S complex was then analyzed using single-particle cryo-EM. After 2D classification, 211,727 ribosomal particles were initially sorted into eight classes, two of which had strong density for eEF3 (I: class 1 and 2), one had density for eEF2 and a weak density for eEF3 (II: class 3), two had weak density for eEF3 (III: class 4 and 5), and the remaining three classes (IV: class 6–8) were poorly defined and had low resolution (Appendix Fig S3). The two classes (I: class 1 and 2) with strong density for eEF3 were combined and a local mask was applied around eEF3 and a second round of focused sorting and refinement was undertaken, eventually yielding a cryo-EM reconstruction of the eEF3-80S complex (Fig 4B and Movie EV1) with an average resolution of 3.3 Å (Appendix Fig S4A). Local resolution calculations reveal that while the majority of the ribosome is well resolved (Appendix Fig S4B and C), the resolution of eEF3 varies, with ribosome-interacting regions being better resolved (3–4 Å) than the peripheral regions (4–6 Å) (Appendix Fig S4D). As observed in the previous eEF3-80S structure (Andersen *et al*, 2006), density for eEF3 spans between the head of the 40S subunit and the central protuberance of the 60S subunit (Fig 4B). We therefore also employed multi-body refinement (Appendix Fig S3), which led to slight improvements in the local resolution for the region of eEF3 that interacts with the head of the 40S subunit (Appendix Fig S4E–H). The resolution allowed an accurate domain-by-domain fit of the previous 2.4 Å X-ray structure of yeast eEF3 (Andersen *et al*, 2006) into the cryo-EM density map of the eEF3-80S complex (Appendix Fig S4A, S5A–C and Movie EV1). Electron density for many side chains could be observed, especially bulky and aromatic amino acids (Appendix Fig S5A–C), enabling us to generate an initial model for the N-terminal HEAT domain, as well as C-terminal ABC1/2 and CD. By contrast, the more peripheral 4HB domain was poorly resolved and was modeled only at the secondary structure level, and the final 68 residues of the C terminus including the TAP-tag had no density and were therefore not included in the model. We note that electron density consistent with ATP (or ADPNP) was observed in the active sites of the ABC1 and ABC2 nucleotide-binding domains (NBDs) (Appendix Fig S5D–K). This is consistent with the closed conformation we observed for ABC1 and ABC2 NBDs, which is similar to the closed conformations adopted in the 70S-bound ABCF protein VmlR from *B. subtilis* (Crowe-McAuliffe *et al*, 2018) and archaeal ABCE1-30S post-splitting complex (Nurenberg-Goloub *et al*, 2020), but distinct from the open conformation observed for the free state of ABCE1 (Barthelme *et al*, 2011) (Fig EV3A–D). Indeed, modeling of an open conformation of eEF3 on the ribosome suggests that it is incompatible with stable binding, leading to either a clash of the HEAT repeat region and the ABC2-CD with the 40S head and the central protuberance of the 60S subunit, respectively, or dissociation of the ABC1-HEAT-4HB domain from

the 40S subunit (Fig EV3E–J). These observations support the idea that eEF3 binds to the ribosome in the "closed" ATP-bound conformation and that ribosome-stimulated hydrolysis of ATP to ADP, with the associated conformational change into the open conformation, would destabilize eEF3 binding and promote dissociation (Chakraburtty, 1999).

### Interaction of eEF3 on the yeast 80S ribosome

The overall geometry of eEF3 on the 80S ribosome is similar to that reported previously for eEF3 (Andersen *et al*, 2006), as well as its homologue in yeast New1p (Kasari *et al*, 2019b), namely oriented such that the N-terminal HEAT interacts with the 40S subunit, whereas the C-terminal CD contacts exclusively the 60S subunit (Fig 4C–E). With the improved resolution compared to the previous eEF3-80S structure (Andersen *et al*, 2006), details of the interactions of eEF3 with the ribosomal components can now be more accurately described. As with New1p (Kasari *et al*, 2019b), the majority of the contacts with the 40S subunit are established by the HEAT domain of eEF3 with components of 40S head including the tip of ES39S of the 18S rRNA as well as r-proteins uS13 and uS19 (Figs 4D and E, and EV4A–D). The majority of the interactions involve positively charged residues located within the loops linking the HEAT repeat α-helices, such as Lys101 (loop α3-α4), Lys141 (loop α5-α6), and Lys182 (loop α7-α8) that are in close proximity to the nucleotides A1360 and U1361 of ES39S (Fig EV4C). Interactions are also observed between the HEAT domain loops (Gln143, Asp180, Glu184) with charged residues (Gln25, Arg130, Asp127, and Arg134) of eS19 (Fig EV4C). The distal end of the HEAT domain (conserved residues 302–307, Glu304, Arg306, Glu307) also establishes interactions with uS13 in the vicinity of His78 and Arg80 (Fig EV4D). The 4HB and ABC1 domains of eEF3 do not interact with any ribosomal components, whereas ABC2 bridges the inter-subunit space by forming protein–protein interactions with uS13 on the 40S involving Arg871 and Arg872 (Fig EV4D) and uL5 and uL18 on the 60S subunit (Fig EV4E and F). In addition, the loop between the β2 and β3 strands of the CD contains positively charged residues (Arg802, Arg803, Lys804, Lys806, and Asn807) that appear to interact with nucleotides U32-G41 of the 5S rRNA (Fig EV4F).

### eEF3 facilitates E-site tRNA release by opening the L1 stalk

To address which functional states of the ribosome are stably bound by eEF3, we implemented additional rounds of 3D classification of the classes with strong density for eEF3 (group I: classes 1 and 2), but with a focus on the L1-stalk and tRNAs (Appendix Fig S3). This yielded four subclasses (Ia-Id) that all exhibited strong density for eEF3, but differed with respect to their functional state. Of these four classes, two pre-translocation (PRE) states were observed that could be refined to 3.5 and 4.0 Å, respectively (Fig 5A and B). PRE-1 contained classical A- and P-site tRNAs, whereas PRE-2 contained a classical A-site tRNA and a hybrid P/E-site tRNA. In both PRE-1 and PRE-2, density for the nascent polypeptide chain could be observed extending from the A-site tRNA and entering into the ribosomal exit tunnel (Fig 5A and B). Since eEF3 has been suggested to play a role in the dissociation of E-site tRNA from the ribosome, the observation of eEF3 bound to various PRE-state ribosomes was surprising and the implication of this with respect to dissociation of eEF3 from

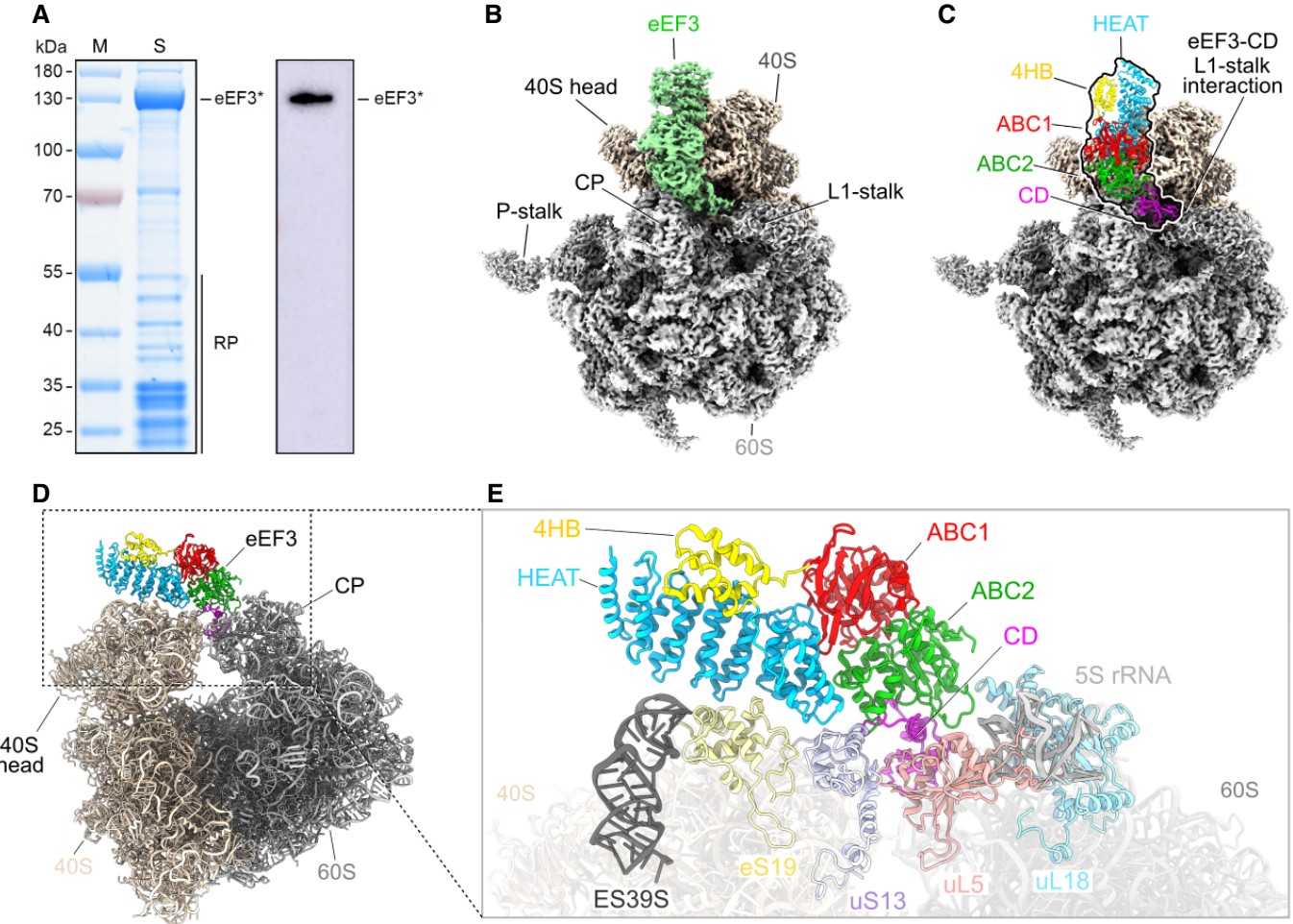

**Figure 4. eEF3 in complex with the *Saccharomyces cerevisiae* 80S ribosome.**

A   SDS gel and Western blot of the native pull-out from *S. cerevisiae* TAP-tagged eEF3. eEF3* labels the TEV cleavage product of eEF3-TAP, which still carries the calmodulin binding peptide of the TAP-tag and which is recognized by the antibody. RP, ribosomal proteins.

B   Cryo-EM reconstruction of the native eEF3-80S complex with segmented densities for the 60S (gray), the 40S (tan), and eEF3 (pastel green).

C   Fitted eEF3 molecular model colored according to its domain architecture, HEAT (blue), 4HB (yellow), ABC1 (red), ABC2 (green), and CD (magenta).

D, E   (D) Overview on the eEF3-80S molecular model and (E) zoom illustrating eEF3 and its ribosomal interaction partners. ES39S (dark gray), eS19 (pale yellow), uS13 (pastel violet), uL5 (coral), uL18 (light blue), 5S rRNA (light gray).

the ribosome will be discussed later where we focus on the identification of PRE-3, PRE-4, and POST-1 states that lack clear density for eEF3. In addition to the PRE-state complexes, two post-translocation (POST) state ribosomes were obtained, POST-2 and POST-3, which could be refined to 3.9 and 3.8 Å, respectively (Fig 5C and D). POST-2 contained classical P- and E-site tRNAs, whereas POST-3 contained a P-site tRNA, but lacked an E-site tRNA (Fig 5C and D). In both the POST-2 and POST-3 states, density for the nascent polypeptide chain could also be visualized attached to the P-site tRNA (Fig 5C and D). The presence and absence of E-site tRNA in the POST-2 and POST-3, respectively, suggested that these complexes could represent the states before and after eEF3-mediated E-site tRNA release, respectively. Consistent with this hypothesis, the L1 stalk is observed in a closed conformation in POST-2 that would prohibit release of the E-site tRNA from the ribosome. We term this closed conformation L1-"int" for intermediate since it is

not as closed as the L1-"in" conformation observed in the classical POST-1 state (Fig EV5A–D). By contrast, in POST-3 the L1 stalk has moved into an open or L1-"out" conformation that would enable E-site tRNA dissociation (Fig 5E and F). In POST-2, we observe no direct interaction between eEF3 and the E-site tRNA, but rather between the CD of eEF3 and the L1 stalk (Fig 5G and H). Specifically, the contact between eEF3 and the L1 stalk is from the tip of the CD with domain II of the uL1 protein (Fig 5H). Unfortunately, the L1 region is poorly ordered due to high flexibility, nevertheless, it appears that the interactions are dominated by charge complementarity between the conserved glutamates (Glu790, Glu819 and Glu826) in the CD that are in close proximity to a highly conserved lysine-rich region (Lys91-92, Lys95, Lys97-98, Lys101-102, Lys105-106) present in domain II of uL1 (Fig 5H). The large number of lysine residues in this region of uL1 is intriguing since CD are best known from chromatin remodeling proteins where they bind to

methylated lysines (me-K) present in histone tails (Nielsen *et al*, 2002; Taylor *et al*, 2007; Yap & Zhou, 2011). Methylation of few non-histone proteins has been reported, one of which is the uL1 protein that is methylated on a single lysine (me-K46) (Webb *et al*, 2011); however, K46 is located in domain I of uL1, far from the CD interaction site (Fig EV5E and F). Moreover, the hydrophobic pocket that is generally formed by three aromatic residues to interact with the me-K, does not appear to be present in eEF3 (Fig EV5G–I). Collectively, our findings imply that eEF3 is likely to facilitate E-site tRNA release (as we observe biochemically, Fig 2D) indirectly by promoting the transition of the L1 stalk from the "in" to the "out" conformation (Fig 5E and F), rather than from directly using the CD to dislodge the E-site tRNA from its binding site.

## eEF3 binds stably to non-rotated ribosomal states

We noted that although eEF3 bound stably to a mixture of different PRE and POST states (Fig 5A–D), these ribosomal states nevertheless had a common feature in that they are all non-rotated, i.e., they have a non-rotated 40S with respect to the 60S subunit. This non-rotated state is critical for eEF3 to establish simultaneous interactions with the ES39S, uS13 and eS19 on the head of the 40S as well as the 5S rRNA, uL5 and uL18 on the 60S subunit, as described for the high-resolution eEF3-80S complex (Figs 4D and E, and EV4), thereby providing an explanation for the strong density observed for eEF3 in the non-rotated PRE-1/-2 and POST-2/-3 state. To explore this idea further, we also analyzed the classes where the density for eEF3 was poorly resolved (Appendix Fig S3, group II and III). Refinement of group II (class 3) to an average resolution of 4.1 Å yielded an eEF2-bound ribosome with chimeric ap/P- and pe/E-tRNAs (POST-1) and poorly defined density for eEF3 (Fig 6A). In addition to intersubunit rotation by 8.5°, we also observed a counter-clockwise (toward the E site) head swivel of 20° (Fig 6B). As expected, the rotation and head swivel movements have a large impact on the eEF3-binding site, namely moving the 40S interaction partners, such as ES39S, eS19, and uS13, by 16 Å for the rRNA, 17 Å for eS19 up to 27 Å for uS13 relative to the 60S and their position in the non-rotated ribosome (e.g., POST-3) (Appendix Fig S6A and B). Closer inspection of the density for eEF3 in POST-1 indeed revealed that the majority of the density remains associated with the

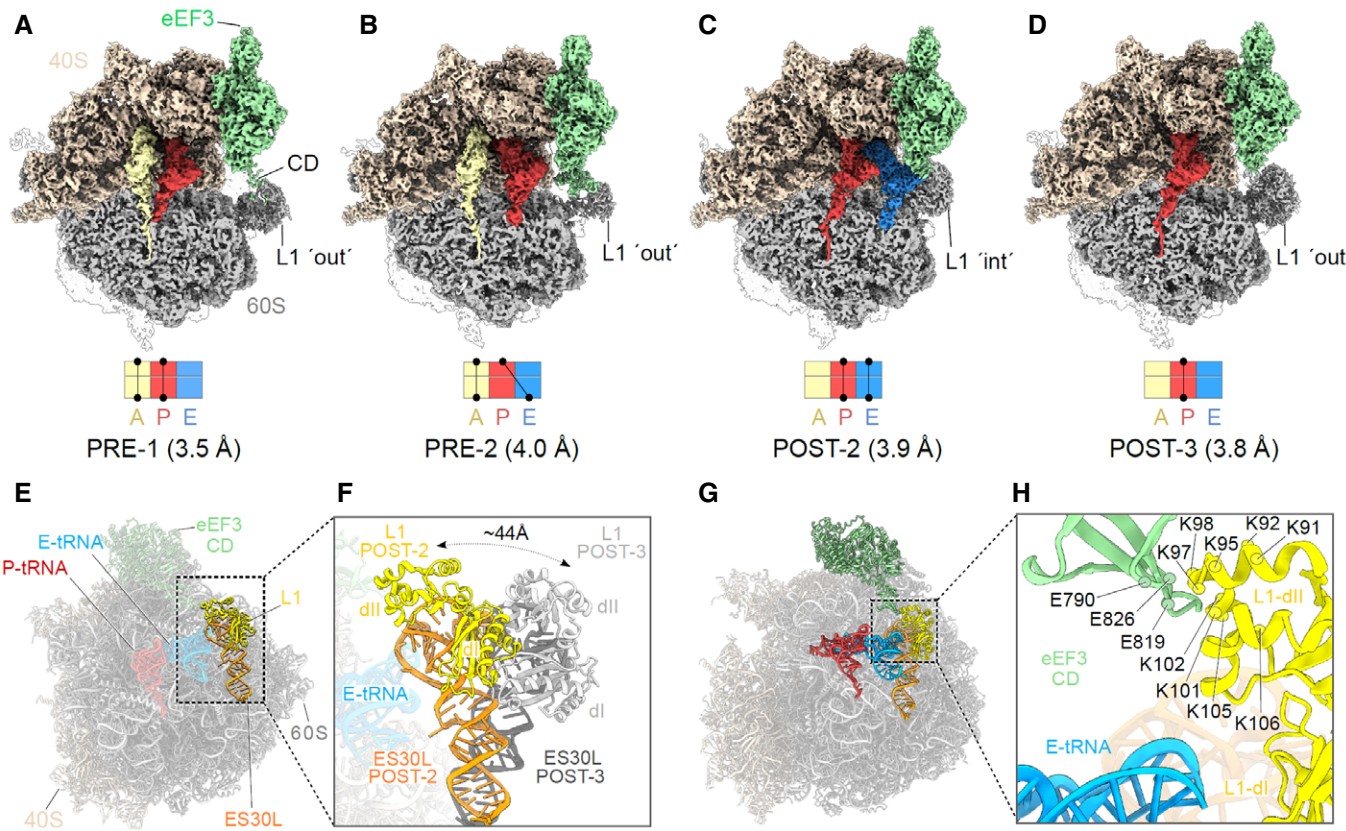

**Figure 5. eEF3 bound to a non-rotated 80S ribosome.**

A–D Cryo-EM maps of eEF3 bound to non-rotated ribosomal species with isolated densities for the 60S, 40S and eEF3 (colored as in Fig 1), as well as the A- (pale yellow), P- (red), and E-tRNA (blue). The eEF3 ligand is binding the ribosome in the pre-translocational (PRE) state termed as (A) PRE-1 (A/A-, P/P-tRNA) and (B) PRE-2 (A/A-, P/E-tRNA) as well as to the post-translocational (POST) version named (C) POST-2 (P/P-, E/E-tRNA) and (D) POST-3 (P/P-tRNA).

E, F (E) Overview of the POST-2 eEF3-80S molecular model highlighting the L1 protein (yellow) and ES30L (orange) and its (F) zoom depicting the magnitude of the L1-stalk movement from the POST-2 state (L1-stalk' int') to the POST-3 state (L1-stalk'out'). L1-POST3 (light gray), ES30L-POST3 (dark gray).

G Different view of the model shown in (E) highlighting eEF3, L1, ES30L as well as the P/P- and E/E-tRNA.

H Zoom of (G) showing the contact of the eEF3-CD and the L1 protein as well as the distance of both to the E-tRNA. The residues are displayed as circles and labeled.

60S and that both rotation and swiveling causes a loss of connectivity between eEF3 and the head of the 40S (Appendix Fig S6C). Since there is defined density for eEF3 in the non-rotated POST-2 and POST-3, but eEF3 is disordered in the eEF2-bound rotated POST-1 state, this indicates that stable binding of eEF3 occurs during or subsequent to eEF2 dissociation from the ribosome when the 40S resets and the ribosome returns to a non-rotated and non-swiveled conformation. We suggest that eEF3 facilitates late stages of translocation by stabilizing the non-rotated state after eEF2 has promoted the movement of the tRNA–mRNA complex (Fig 1D).

## Intersubunit rotation leads to unstable eEF3 binding on the ribosome

Further sorting and refinement of group III (Appendix Fig S3, classes 4 and 5) yielded two distinct PRE states, namely PRE-3 containing a classical A-site tRNA and a hybrid P/E-site tRNA, and PRE-4

containing hybrid A/P- and P/E-site tRNAs (Fig 6C and D), which could be refined to average resolutions of 4.2 and 3.3 Å, respectively. Analogous to PRE-1 and PRE-2, both PRE-3 and PRE-4 states also had density for the nascent polypeptide chain extending from the A-site tRNA into the ribosomal exit tunnel (Fig 6C and D). However, unlike PRE-1 and PRE-2, the 40S in PRE-3 and PRE-4 was rotated by 9.6° and 11° relative to the 60S subunit (Fig 6E and F). In the PRE-3 state, little to no density for eEF3 was observed (Fig 6C), suggesting that the factor had dissociated from the ribosome during sample and/or grid preparation. While significant density was observed for eEF3 in PRE-4, the density was poorly resolved (Fig 6D). Analogous to POST-1, the rotation observed in PRE-3 and PRE-4 appears to also disrupt the interactions with the 40S subunit such that the majority of the eEF3 density is connected to the 60S subunit (Appendix Fig S6D–I). Collectively, this supports the suggestion that stable binding of eEF3 to the ribosome requires a non-rotated state and that intersubunit rotation leads to

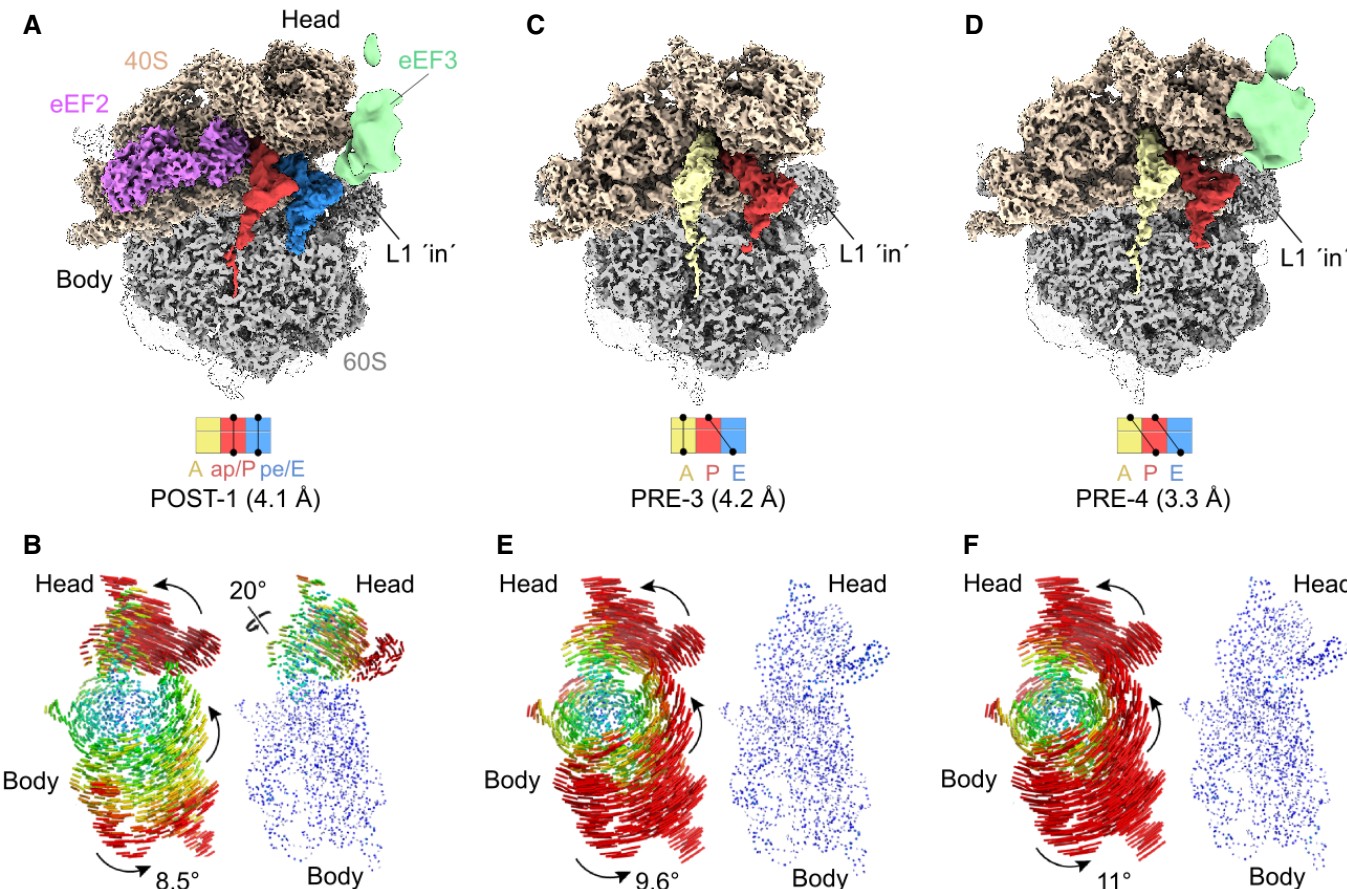

**Figure 6. Rotated *Saccharomyces cerevisiae* ribosome partially bound by disordered eEF3 ligand.**

A   Cryo-EM reconstruction of the eEF2 bound 80S ribosome adopting a rotated POST-1 state bearing chimeric ap/P- and pe/E-tRNA as well as a disordered eEF3.
B   Subunit rotation and head swivel observed in the *S. cerevisiae* eEF3-80S complex derived from the selected class from the 3D classification shown in (A). 18S rRNA structures illustrating the degree of rotation of the small subunit relative to the large subunit based alignments of the large subunit from a non-rotated structure reference (PDB ID: 6SNT) (Matsuo *et al*, 2020). The degree of head swiveling (right side) is illustrated based on alignments of the 18S rRNA body from a non-headed swivel reference structure (PDB ID: 6SNT) (Matsuo *et al*, 2020).
C, D   80S ribosome in rotated (C) PRE-3 state containing an A/A- and P/E-tRNA and (D) PRE-4 state bearing the hybrid A/P- and P/E-tRNA and bound by the disordered eEF3 ligand.
E, F   Representation of the rotation of the subunit and the 40S-head swivel shown for volumes depicted in (C) and (D), respectively.

destabilization of eEF3 binding (Fig 6C and Appendix Fig S6F). eEF3 remains stably bound to the ribosome subsequent to E-site tRNA release and is still stably bound when the next aminoacyl-tRNA has accommodated into the A site and undergone peptide bond formation. Rather it appears that the critical step that leads to conformational rearrangements in eEF3 is the transition from non-rotated to rotated states. In this respect, it is interesting to note that PRE-3 has the same tRNA arrangement as PRE-2, namely both containing a classical A-site tRNA with nascent chain and a deacy-lated hybrid P/E-site tRNA, yet adopt completely different rotational states (Figs 5B and 6C). This is somewhat surprising since forma-tion of hybrid states in higher eukaryotes is generally assumed to be concomitant with intersubunit rotation (Budkevich *et al*, 2011; Svidritskiy *et al*, 2014; Behrmann *et al*, 2015); however, we note that subunit rotation and tRNA movement is only loosely coupled on bacterial ribosomes (Fischer *et al*, 2010). Since rotation of PRE-2 would result in a state similar to PRE-3, we suggest that subunit rotation is what triggers the low affinity form of eEF3, effectively leading to unstable ribosome binding.

## Discussion

Collectively, our findings suggest a model for the role of eEF3 during the translocation elongation cycle (Fig 7A–G and Movie EV2). The ability to capture four different PRE and three different POST states of the ribosome by co-immunoprecipitation of tagged eEF3 suggests that eEF3 is omnipresent during translation elonga-tion in yeast, which is consistent with the similar copy number between eEF3 and ribosomes (Firczuk *et al*, 2013). Our biochemi-cal data show that the main function of eEF3 is to accelerate the E-site tRNA release from the ribosome during late stages of mRNA-tRNA translocation (Figs 1 and 2). These *in vitro* findings are in strong agreement with the *in vivo* ribosome profiling exper-iments. Similar to previous studies where eEF3 was depleted in the cell (Kasari *et al*, 2019a), we observe general defects in trans-lation elongation. However, our application of high-resolution ribosome profiling, using combinations of elongation inhibitors for library preparation, allowed us to follow ribosomes trapped in distinct functional states and thereby identify the specific elonga-tion defect (Fig 3). In particular, we find that eEF3 depletion enriched ribosomes trapped in a pre-translocation state (28 nt RPFs), fully consistent with the observation that eEF3 promotes a late step in translocation that depends on E-site tRNA release. A strong prediction of the biochemical and ribosome profiling results is that the ribosomes that accumulate in cells on eEF3 depletion would contain three separate tRNAs.

Cryo-EM structures suggest two mechanisms by which eEF3 can facilitate translocation, namely by favoring the transition toward the non-rotated conformation of the ribosome and by influencing the conformation of the L1 stalk. As in all kingdoms of life, yeast eEF2 stabilizes the rotated state of the ribosome and accelerates the movement of the peptidyl-tRNA from the A to the P site (Fig 7D and E). In bacteria, the eEF2 homolog EF-G promotes the small subunit head swiveling, which allows the mRNA–tRNAs to move relative to the large subunit (Zhou *et al*, 2014; Belardinelli *et al*, 2016; Wasser-man *et al*, 2016). Resetting the swiveled conformation triggers the release of EF-G and the E-site tRNA, which occurs rapidly on a

millisecond time scale (Belardinelli *et al*, 2016), and does not require auxiliary factors. The present structures suggest which features of the mechanism are conserved in yeast. Similar to EF-G, eEF2 catalyzes head swiveling and mRNA–tRNA translocation (Fig 7E), whereas dissociation of the factor and back-swivel occur in a later step (Fig 7E and F). The cryo-EM observation of discrete POST-1 and POST-2 states suggests that this transition is relatively slow in yeast cells, even in the presence of eEF3. Stabilization by eEF3 of the non-rotated state may accelerate the reaction and contri-bute to the directionality of translocation. Since we do not observe any difference in the association of eEF2 with ribosomes upon eEF3 depletion (Fig EV2D), we do not think that eEF3 plays an important role for eEF2 dissociation. Similarly, small differences in the rates of peptide bond formation (Figs 1D and 2B) can be attributed to the effect of eEF3 on the ribosome dynamics, but do not have a major effect *in vivo* (Fig 3D, G and H). By contrast, we observe dramatic differences in the rates of E-site tRNA release in the presence and absence of eEF3 (Fig 2C–E).

In contrast to bacterial ribosomes, which release the E-site tRNA quickly (Uemura *et al*, 2010; Belardinelli *et al*, 2016), dissociation of deacylated tRNA from yeast ribosome in the absence of eEF3 is very slow, occurring in the seconds range (this paper and (Garreau de Loubresse *et al*, 2014)). Even in the presence of eEF3, E-site tRNA release is observed as a separate step from POST-2 to POST-3 state (Fig 7F and G). In our POST-2 state, the CD of eEF3 is seen to directly contact the L1 stalk, but not the E-site tRNA, suggesting that dissoci-ation of the E-site tRNA by eEF3 is facilitated by shifting the L1 stalk from an "int" to an "out" conformation (Fig 7F and G), as was hypothesized previously (Triana-Alonso *et al*, 1995; Andersen *et al*, 2006). We note, however, that we do not observe release of the E-site tRNA to be dependent on binding of the ternary complex to the A site (Fig 2C), as proposed previously (Triana-Alonso *et al*, 1995). We also do not observe the presence of eEF1A in any of our cryo-EM states, and depletion of eEF3 does not lead to accumulation of eEF1A (or eEF2) on the ribosome (Fig EV2D). Rather, our *in vivo* eEF3-depletion studies suggest that the ribosomes become blocked with all three tRNA binding sites being occupied (Fig 7H). As biochemical experiments show that peptidyl transfer can occur in the absence of eEF3 (Fig 1D), we envisage that ribosomes are trapped in the POST-2 state with deacylated tRNAs in the E and P sites and a peptidyl-tRNA in the A site. Importantly, the subsequent translocation step would not be possible, probably because the presence of deacylated tRNA at the E site inhibits the ribosome rotation and tRNA move-ment. Such a state with three tRNAs on the ribosome would be consistent with the loss of 21 nt RPFs in the ribosome profiling experiments observed when eEF3 is depleted from the cells (Figs 3B and D). While it has been shown by smFRET that bacterial ribo-somes with three tRNAs assume a partially rotated state due to slow kinetics of E-site tRNA release which hinders translocation (Choi & Puglisi, 2017), it is tempting to speculate that fungi may have evolved to utilize eEF3 to overcome this kinetic hurdle, promoting rapid translation.

eEF3 has a marked binding preference for the non-rotated states of the ribosome, e.g., PRE-1 and PRE-2, as well as POST-2 and POST-3 states, and becomes destabilized in the rotated PRE-3 and PRE-4 that arise when the A- and/or P-site tRNAs move into hybrid sites (Fig 7C and D). Similarly, eEF3 does not stably interact with the eEF2-bound POST-1 state bearing chimeric ap/P- and pe/E-

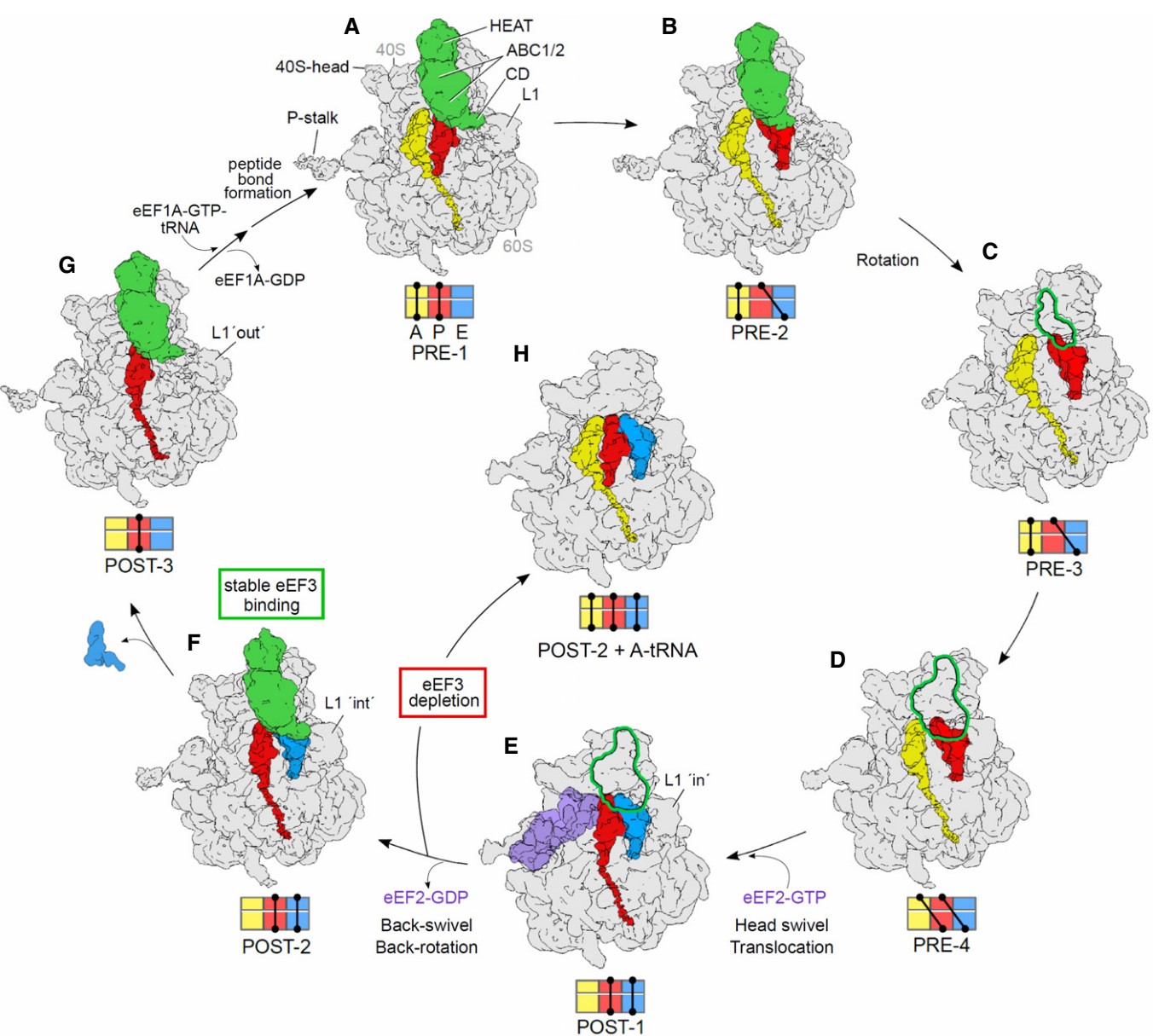

**Figure 7. Functional role of eEF3 in the framework of the elongation cycle.**

A, B    eEF3 binds to (A) non-rotated 80S in a pre-translocation (PRE) state bearing classical A/A- and P/P-tRNA (PRE-1) as well as to (B) non-rotated state occupied by A/A- and a hybrid P/E-tRNA (PRE-2).

C, D    (C) Rotation of the ribosomal subunits leads to unstable binding of eEF3 to a ribosome with A/A- and P/E-tRNA (PRE-3) as well as to (D) a fully rotated ribosomal species bearing hybrid A/P- and P/E-tRNAs (PRE-4).

E    Binding of eEF2-GTP facilitates 40S-head swiveling and translocation of the hybrid A/P- and P/E-tRNA into the chimeric ap/P and pe/E-tRNA positions (POST-1).

F    After dissociation of eEF2-GDP and the resulting back-swivel and back-rotation of the ribosome, eEF3 binds stably to the non-rotated ribosome with a classical P/P- and E/E-tRNA (POST-2) with L1-stalk in the "int" position.

G    The eEF3-CD directly interacts with L1 stabilizing an "out" conformation that facilitates release of the E-site tRNA. eEF3 remains bound to the non-rotated ribosome bearing a P/P-tRNA (POST-3). Transition from states E to F and F to G are accelerated by eEF3. Binding of eEF1a-ATP-tRNA ternary complex and subsequent peptide bond formation results in formation of PRE-1, as seen in (A).

H    In the absence of eEF3, we suggest that POST-2 ribosomes formed after eEF2 dissociation can bind an A-tRNA, but cannot translocate further because of the presence of three tRNAs on the ribosome.

Data information: Volumes (A–G) are representing the cryo-EM reconstructions from the eEF3-TAP pull-out, whereas volume (H) shows a potential scenario based of the results of the eEF3-depletion studies. All maps are filtered to 6 Å. The outline shown in (C–E) assigns the disordered eEF3 ligand present in these volumes.

tRNAs (Fig 7E, Appendix Fig S6C). Comparison of rotated and non-rotated ribosome conformations reveals that subunit rotation disturbs the interactions between eEF3 and the 40S subunit,

providing a rationale for the destabilization (Appendix Fig S6). By contrast, the back-rotation and back-swivel of the head that accompanies eEF2 dissociation yields a classical non-rotated POST-2 state

with P- and E-site tRNAs that is an optimal substrate for stable binding of eEF3 (Fig 7F). In this state, we observe that eEF3 adopts a closed conformation, presumably with two molecules of ATP (or ADPNP) bound within the ABC1 and ABC2 NBDs (Appendix Fig S5D–K). The presence of ADPNP prevents eEF3-mediated E-site tRNA release, suggesting that ATP hydrolysis in at least one of the NBD of eEF3 is necessary for this process (Triana-Alonso et al, 1995). Although eEF3 appears to adopt a closed conformation in the POST-3 state following E-site tRNA release (Fig 7G), we cannot rule out that in the cell eEF3 adopts a partially open hybrid ATP/ADP state and that during the last purification step performed in the presence of ADPNP, the ADP is displaced by ADPNP, thereby reverting eEF3 from the partially closed into a fully closed conformation. If this were the case, one could speculate that hydrolysis of one ATP to ADP occurs concomitant with E-tRNA release when transitioning from POST-2 to POST-3 (Fig 7F and G) and hydrolysis of the second ATP to ADP may be stimulated by subunit rotation, explaining the destabilizing of eEF3 on the ribosome (Fig 7B and C). In the latter case, even if the ADPs are replaced with ADPNP leading to a closed conformation, the rotated state precludes stable binding of eEF3, although we cannot rule out that PRE-3, PRE-4, and POST-1 (Fig 7C–E) is stably bound by eEF3 in a partially closed state. Finally, the presence of eEF3 on the ribosome during decoding, e.g., at the transition from POST-3 to PRE-1 state may affect the relative rates of decoding reactions, which may explain the effect of eEF3 on translation fidelity (Uritani & Miyazaki, 1988) and account for the observed genetic interactions between eEF1A and eEF3 (Anand et al, 2003; Anand et al, 2006).

In addition to providing insight into the function of eEF3, our study fills the gap for various missing structures of the yeast translation elongation cycle (Fig 7 and Movie EV2). While yeast PRE-1 (Fig 7A) and PRE-4 (Fig 7D) states have been reported recently in yeast (Ikeuchi et al, 2019; Buschauer et al, 2020; Matsuo et al, 2020), those structures did not contain eEF3. We also observe the novel non-rotated PRE-2 (Fig 7B) as well as canonical rotated PRE-3 states (Fig 7C). PRE-1, PRE-3, and PRE-4 are similar to classical and rotated states observed in mammalian systems (Budkevich et al, 2011; Behrmann et al, 2015). While there have been multiple structures of yeast eEF2 on the ribosome (Gomez-Lorenzo et al, 2000; Spahn et al, 2004; Taylor et al, 2007; Sengupta et al, 2008; Pellegrino et al, 2018), none contained the physiological configuration of two tRNAs being translocated by eEF2, as observed in our POST-1 state (Fig 7E). We note however that similar functional states to POST-1 have been observed in rabbit and human systems (Behrmann et al, 2015; Flis et al, 2018). Finally, although an in vitro reconstituted POST-3 state with eEF3 (Fig 7G) has been reported at 9.9 Å resolution (Andersen et al, 2006), here we present a native POST-3 at 3.8 Å as well as additionally capturing POST-2 state with eEF3 and E-site tRNA (Fig 7F).

Collectively, our biochemical, genetic, and structural analysis sheds light into the critical function of eEF3 during translation in yeast. Ultimately, rapid translation in yeast requires fast movement of the tRNAs through the ribosome and their release from the E site. Here, we show that eEF3 fulfills both roles by facilitating late steps in tRNA translocation and by inducing the L1 stalk to adopt an out conformation allowing E-site tRNA release. Some questions for the future are to understand the role of ATP hydrolysis in eEF3 function and whether it is linked to E-site tRNA release and/or the stability

of eEF3 binding, as well as why a factor such as eEF3 that is essential for translation yeast does not appear to have a homolog in higher eukaryotes, such as humans.

# Materials and Methods

### E. coli-competent cells and growth conditions

eIF1 was cloned into a pET22b plasmid without affinity tag. eIF1A, eIF5, eIF5A, and eIF5B-397C (lacking 396 amino acids from the C-termini) were cloned into a pGEX-6P1 plasmid with a GST tag. Recombinant proteins were expressed in E. coli BL21(DE3) cells from IPTG-inducible plasmids at 37°C for eIF1, eIF5A, and eIF5B-397C. eIF1A and eIF5 were expressed from IPTG-inducible plasmids at 16°C.

### Saccharomyces cerevisiae strain and growth conditions

The S. cerevisiae cells for ribosomal subunits, eIF2, eEF1A, eEF2, and eEF3, were grown in a 250 l bioreactor in 1xYPD (20 g/l peptone, 10 g/l yeast extract, 20 g/l glucose) at 30°C. The S. cerevisiae cells for eIF3 were grown in a 250 l bioreactor in CSM-LEU-URA media at 30°C. The S. cerevisiae eEF3-TAP strain (GE Healthcare/Dharmacon, Strain: BY4741, GeneID: 850951) was grown to $OD_{600}$ = 0.8 in an InnovaX large-capacity incubator shaker in 1xYPD at 30°C.

For ribosome profiling, WT (BY4741) and yCW35 (MATa his3Δ1 leu2Δ0 met15Δ0 ura3Δ0 hef3::KanMX HO::adh1p-osTIR1-Ura3 SkHIS3-GAL1p-yef3-3mAID::NatMX) strains were grown in YPGR (20 g/l peptone, 10 g/l yeast extract, 20 g/l galactose + 20 g/l raffinose) overnight at 30°C, collected by centrifugation, washed, and resuspended in YPD at OD 0.003 and 0.02, respectively. After 8 h, cells were harvested at OD 0.55 and 0.45, respectively. Cells were harvested by fast filtration and flash frozen in liquid nitrogen.

### Buffers

The following buffers were used for purification of translation components:

- Buffer 1. 20 mM HEPES/KOH pH 7.5, 100 mM KOAc, 2.5 mM Mg(OAc)$_2$, 1 mg/ml heparin sodium salt, 2 mM DTT.
- Buffer 2. 20 mM HEPES/KOH pH 7.5, 100 mM KOAc, 400 mM KCl, 2.5 mM Mg(OAc)$_2$, 1 M sucrose, 2 mM DTT.
- Buffer 3. 20 mM HEPES/KOH pH 7.5, 100 mM KOAc, 400 mM KCl, 2.5 mM Mg(OAc)$_2$, 1 mg/ml heparin sodium salt, 2 mM DTT.
- Buffer 4. 50 mM HEPES-KOH pH 7.5, 500 mM KCl, 2 mM MgCl$_2$, 2 mM DTT.
- Buffer 5. 50 mM HEPES-KOH pH 7.5, 500 mM KCl, 5 mM MgCl$_2$, 2 mM DTT, 0.1 mM EDTA, 5% sucrose.
- Buffer 6. 50 mM HEPES-KOH pH 7.5, 500 mM KCl, 5 mM MgCl$_2$, 2 mM DTT, 0.1 mM EDTA, 30% sucrose.
- Buffer 7. 50 mM HEPES-KOH, 100 mM KCl, 250 mM sucrose, 2.5 mM MgCl$_2$, 2 mM DTT.
- Buffer 8. 20 mM HEPES-NaOH pH 7.5, 150 mM NaCl, 5% glycerol, 4 mM β-mercaptoethanol (β-me).

- Buffer 9. Same as buffer 8 with 1 M NaCl.
- Buffer 10. Same as buffer 8 without NaCl and glycerol.
- Buffer 11. 20 mM HEPES-KOH pH 7.5, 200 mM KCl, 2 mM DTT.
- Buffer 12. 20 mM HEPES-NaOH pH 7.5, 500 mM NaCl, 5% glycerol, 4 mM β-me.
- Buffer 13. 50 mM HEPES-NaOH pH 7.5, 400 mM NaCl, 5% glycerol, 4 mM β-me.
- Buffer 14. 20 mM HEPES-KOH pH 7.5, 500 mM KCl, 10% glycerol, 3 mM β-me.
- Buffer 15. 20 mM 3-(N-morpholino)propane sulfonic acid (MOPS)-KOH pH 6.7, 500 mM KCl, 10% glycerol, 3 mM β-me.
- Buffer 16. 20 mM HEPES-NaOH pH 7.5, 500 mM NaCl, 30 mM L-glutathione reduced, 5% glycerol, 4 mM β-me.
- Buffer 17. 20 mM HEPES-KOH pH 7.5, 100 mM KCl, 2 mM DTT.
- Buffer 18. 20 mM HEPES-KOH pH 7.5, 200 mM KCl, 2 mM DTT.
- Buffer 19. 20 mM HEPES-KOH pH 7.5, 100 mM KCl, 5% glycerol, 2 mM DTT.
- Buffer 20. 20 mM HEPES-KOH pH 7.5, 100 mM KCl, 5% glycerol, 2 mM DTT.
- Buffer 21. 20 mM HEPES-NaOH pH 7.5, 1 M NaCl, 5% glycerol, 4 mM β-me.
- Buffer 22. 50 mM Tris–HCl pH 7.5, 5 mM $MgCl_2$, 50 mM $NH_4Cl$, 0.2 mM PMSF, 10% glycerol, 0.1 mM EDTA pH 8.0, 1 mM DTT.
- Buffer 23. 20 mM Tris–HCl pH 7.5, 50 mM KCl, 0.1 mM EDTA pH 8.0, 0.2 mM PMSF, 25% glycerol, 1 mM DTT.
- Buffer 24. 20 mM Tris–HCl pH 7.5, 500 mM KCl, 0.1 mM EDTA pH 8.0, 0.2 mM PMSF, 25% glycerol, 1 mM DTT.
- Buffer 25. 20 mM Tris–HCl pH 7.5, 0.1 mM EDTA pH 8.0, 200 mM KCl, 25% glycerol, 1mM DTT.
- Buffer 26. 20 mM HEPES-KOH pH 7.5, 500 mM KCl, 20 mM imidazole, 10% glycerol, 2 mM β-me.
- Buffer 27. 20 mM HEPES- KOH pH 7.5, 100 mM KCl, 250 mM imidazole, 10% glycerol, 2 mM β-me.
- Buffer 28. 20 mM HEPES- KOH pH 7.5, 100 mM KCl, 10% glycerol, 2 mM DTT.
- Buffer 29. 40 mM Tris–HCl pH 7.5, 15 mM $MgCl_2$, 2 mM spermidine, 10 mM NaCl.
- Buffer 30. 20 mM HEPES-KOH (pH 7.4). 100 mM KOAc, 10 mM Mg(OAc)$_2$, 1 mM DTT.

## Preparation of ribosomal subunits, initiation, and elongation factors

80S ribosomal subunits, eIF2, eIF3, eEF2, and eEF3, were prepared from *S. cerevisiae* as described previously (Pavitt *et al*, 1998; Phan *et al*, 2001; Algire *et al*, 2002; Jorgensen *et al*, 2002; Acker *et al*, 2007; Sasikumar & Kinzy, 2014).

For 80S ribosomal subunits purification, cells were harvested in mid-log phase and resuspended in 1 ml/g of cells in lysis buffer 1. Cell pellets frozen in liquid nitrogen were ground using an ultra-centrifugal mill according to the CryoMill protocol (Retsch©). The lysate was thawed at 4°C, 100 μl DNase, and one EDTA-free protease inhibitor tablet was added and incubated at 4°C for 30 min. The thawed lysate was clarified by centrifugation at 13,000 rpm at 4°C for 30 min. The salt concentration of the supernatant was increased to 500 mM KCl and was then filtered using 1 μm glass fiber filters. Ribosomes in the supernatant were collected in buffer 2 at 45,000 rpm at 4°C for 2 h in a Ti45 rotor.

Ribosomal pellets were resuspended in resuspension buffer 3 and were incubated on ice for 15 min. Ribosomes were collected once more through buffer 2 at 100,000 rpm at 4°C for 30 min in a MLA 130 rotor. The pellets were resuspended in buffer 4, incubated on ice for 15 min with 1 mM puromycin and then 10 min at 37°C. The sample was loaded on a 5–30% sucrose gradient (buffer 5 & 6) and centrifuged at 25,000 rpm at 4°C for 16 h in a Ti32 rotor. The 40S and 60S subunits were collected from gradient fractionation, were exchanged separately into buffer 7, concentrated, and stored at −80°C after being flash frozen in liquid nitrogen.

eIF1 cell pellets was resuspended in buffer 8, lysed in the presence of DNase and protease inhibitor tablet, and purified by HiTrap SP cation exchange chromatography with a linear gradient from 0 to 100% buffer 9 over 60 ml after equilibrating the column and loading eIF1 with buffer 8. The fractions containing eIF1 were pooled and diluted to decrease the salt concentration to 150 mM with buffer 10. Next, eIF1 was purified by HiTrap Heparin chromatography with a linear gradient from 0 to 100% buffer 9 over 60 ml after equilibrating the column and loading eIF1 with buffer 8. The fractions containing eIF1 were pooled and concentrated and further purification was attained by size-exclusion chromatography on a HiLoad 26/600 Superdex 75 pg column with buffer 11. The purified eIF1 protein was concentrated and stored in buffer 5 at −80°C.

eIF1A, eIF5, eIF5A, and eIF5B-397C cell pellets were resuspended in buffer 12, 13, 14, and 15, respectively, and lysed in the presence of DNase and protease inhibitor tablet. All GST-tagged proteins were purified by GSTrap column with 100% buffer 16. After cleavage of the fusion protein with PreScission protease (1 μM final), further purification was attained by size-exclusion chromatography on a HiLoad 26/600 Superdex 75 pg (eIF1A with buffer 17, eIF5 with buffer 18, eIF5A with buffer 19) and HiLoad 26/600 Superdex 200 pg (eIF5B-397C with buffer 20) column. For eIF1A, before size-exclusion chromatography, an additional step of a Resource Q anion exchange chromatography was included after protease cleavage and protein was eluted with 0–100% buffer 21 over 60 ml. The purified proteins were concentrated and stored at −80°C.

eEF1A was purified from *S. cerevisiae*. eEF1A cell pellets was resuspended in buffer 22, lysed using an ultra-centrifugal mill, and the lysate was loaded on a tandom HiTrap Q anion exchange and Hi Trap SP cation exchange column pre-equilibrated in buffer 23. eEF1A was eluted from HiTrap SP with 0–100% buffer 24 over 30 ml. Further purification was attained by size-exclusion chromatography on a HiLoad 26/600 Superdex 200 pg with buffer 25. The purified eEF1A protein was concentrated and stored at −80°C.

### *In vitro* hypusination of eIF5A

eIF5A hypusination enzymes deoxyhypusine synthase (Dys1) and deoxyhypusine hydroxylase (Lia1) were co-expressed in *Escherichia coli* BL21(DE3) cells from IPTG-inducible pQLinkH plasmid with a His6 tag at 37°C. Cell pellet was resuspended in buffer 26, lysed in the presence of DNase and protease inhibitor tablet. The protein was purified by Protino Ni-IDA 2000 affinity chromatography with buffer 27, after equilibrating the column and loading the protein with buffer 26. The eluted protein was dialyzed in buffer 28, concentrated, and stored at −80°C. The *in vitro* hypusination of

eIF5A was performed as described previously (Park *et al*, 2011; Wolff *et al*, 2011).

## Preparation of tRNAs and mRNAs

Initiator tRNA ($tRNA_i^{Met}$) was prepared by *in vitro* transcription using T7 polymerase from a plasmid containing a 92 nucleotides-long DNA with the T7 promoter (underlined) and the initiator tRNA sequence purchased from Eurofins.

5′<u>TAATACGACTCACTATAA</u>GCGCCGTGGCGCAGTGGAAGCGCG CAGGGCTCATAACCCTGATGTCCTCGGATCGAAACCGAGCGGCGC TACCA3′

The DNA was amplified using forward and reverse primers, and the amplified product was *in vitro* transcribed in buffer 29 with 10 mM DTT, 3 mM NTP mix, 0.005 U/µl inorganic Pyrophosphate (PPase), 0.1 U/µl RNase inhibitor, and 0.05 U/µl T7 RNA-polymerase for 4 h at 37°C. Aminoacylation and purification of [$^3$H] Met-tRNA$_i^{Met}$ and [$^3$H]Met-tRNA$_i^{Met}$ (Flu) were performed as described previously (Rodnina *et al*, 1994; Milon *et al*, 2007).

The elongator tRNAs [$^{14}$C]Val-tRNA$^{Val}$, [$^{14}$C]Phe-tRNA$^{Phe}$, and Phe-tRNA$^{Phe}$ were prepared as described in (Rodnina *et al*, 1994). The unstructured 5′ UTR mRNA Met-Phe-Val and mRNA Met-Val-Phe were purchased from IBA (mRNA Met-Phe-Val: 5′GGUC UCUCUCUCUCUCUCU<u>AUG</u>UUUGUUUCUCUCUCUCUC3′ and mRNA Met-Val-Phe: 5′GGUCUCUCUCUCUCUCUCU<u>AUG</u>GUUUUUUCUCUC UCUCUC3′).

## Preparation of initiation and ternary complexes

80S initiation complexes were prepared by incubating 8 µM of eIF2 with 1 mM GTP and 4 µM [$^3$H]Met-tRNA$_i^{Met}$ or [$^3$H]Met-tRNA$_i^{Met}$ (Flu) in YT buffer (30 mM HEPES-KOH pH 7.5, 100 mM KOAc, 3 mM MgCl$_2$) at 26°C for 15 min to form the ternary complex. 2 µM 40S, 10 µM mRNA (uncapped mRNA with unstructured 5′ UTR that alleviates the requirement of eIF4 (Acker *et al*, 2007)), 10 µM eIF-mix (mixture of initiation factors eIF1, eIF1A, eIF3, eIF5), 2 mM DTT, 0.25 mM spermidine, and 1 mM GTP were incubated for 5 min at 26°C before adding 3 µM 60S subunits and 6 µM eIF5B. After incubation, ternary complex was added to the ribosome mixture and the MgCl$_2$ was adjusted to a final concentration of 9 mM before layering on sucrose cushion. ICs were purified by ultracentrifugation through a 1.1 M sucrose cushion in YT9 buffer (30 mM Hepes-KOH pH 7.5, 100 mM KoAc, 9 mM MgCl$_2$), and pellets were dissolved in YT9 buffer.

Ternary complexes eEF1A–GTP–[$^{14}$C]Phe-tRNA$^{Phe}$ and eEF1A–GTP–[$^{14}$C]Val-tRNA$^{Val}$ were prepared by incubating 1 µM eEF1A, 0.1 µM eEF1Bα, 3 mM PEP, 1% PK, 1 mM DTT, 0.5 mM GTP in YT buffer for 15 min at 26°C. 0.2 µM [$^{14}$C]Phe-tRNA$^{Phe}$ or [$^{14}$C]Val-tRNA$^{Val}$ (5 eEF1A:1 aa-tRNA) was added and incubated for additional 5 min at 26°C, followed by addition of 2 µM modified eIF5A.

## Rapid kinetics

Peptide bond formation assay was performed by rapidly mixing initiation complexes (1 µM, 0.25 µM active in tripeptide formation) with the respective ternary complexes as indicated (0.2 µM), eIF5A (2 µM) and eEF3 (4 µM) mixed with 40 µM ATP in a quench-flow apparatus at 26°C. After the desired incubation times, the reactions were quenched by adding KOH to a final concentration of 0.5 M.

Peptides were released by alkaline hydrolysis for 45 min at 37°C. After neutralization with acetic acid, the products were analyzed by HPLC (LiChrospher 100 RP-8 HPLC column, Merck). To form tripeptides, MetPhe-tRNA$^{Phe}$ pre-translocation complexes (0.35 µM) were rapidly mix with the respective ternary complex (0.7 µM) containing eEF2 (1 µM) and eEF3 (4 µM) with ATP (40 µM).

The amount of deacylated tRNA$^{fMet}$(Flu) bound to 80S after translocation was monitored on a BioSuite 450 (Waters) size-exclusion chromatography. 80S 2C (0.35 µM) was incubated without eEF2/ eEF3, or with eEF2 (1 µM) alone, eEF2 (1 µM), and eEF3 (4 µM) together or with eEF2, eEF3, and TC-Val (0.7 µM) for 15 min before applying on a BioSuite 450 gel filtration column. Ribosome complexes and tRNA were eluted using YT buffer (30 mM HEPES-KOH pH 7.5, 100 mM KOAc, 3 mM MgCl$_2$). Fluorescence of tRNA$^{fMet}$(Flu) co-eluting with 80S was monitored using a flow-through fluorescence detector after excitation at 463 nm and emission at 500 nm. Deacylated tRNA release from the E site were performed by rapidly mixing initiation complexes prepared using [$^3$H]Met-tRNA$^{Met}$ (Flu) with the respective ternary complexes as indicated (0.2 µM) and eEF1A (1 µM) in a stopped-flow apparatus (Applied Photophysics) at 26°C. Fluorescein fluorophore was excited at 463 nm, and emission was measured after passing through KV500 long-pass filters (Schott). Experiments were performed by rapidly mixing equal volumes of reactants and monitoring the time courses of fluorescence changes. Time courses depicted in the figures were obtained by averaging 5–7 individual traces.

## Time-resolved Pmn assay

The time-resolved Pmn assay to monitor translocation for the MetPhe-tRNA$^{Phe}$ PRE complex was performed as described previously (Ranjan & Rodnina, 2017). Briefly, PRE complex (0.35 µM) was rapidly mixed with Pmn (2 mM), and/or eEF2 (0.8 µM) and/or eEF3 (2 µM) with ATP (20 µM) in YT buffer at 26°C. Control experiments were carried out with POST complexes prepared by incubating PRE complexes with eEF2 (0.8 µM) and/or eEF3 (2 µM) with ATP (20 µM). POST complexes with MetPhe-tRNA$^{Phe}$ (0.35 µM) in the P site were rapidly mixed with Pmn (2 mM) in a quench-flow apparatus. The reaction was quenched with KOH (0.5 M) and the peptides were released by alkaline hydrolysis for 45 min at 37°C, analyzed by reversed-phase HPLC (LiChrospher 100 RP-8, Merck), and quantified by double-label [$^3$H]Met-[$^{14}$C]Phe radioactivity counting (Wohlgemuth *et al*, 2008).

## Preparation of libraries for yeast ribosome footprints

Preparation of libraries for yeast ribosome footprints was performed as described previously (Wu *et al*, 2019a). Cell pellets were ground with 1 ml yeast footprint lysis buffer [20 mM Tris–Cl (pH 8.0), 140 mM KCl, 1.5 mM MgCl$_2$, 1% Triton X-100 with specified elongation inhibitors] in a Spex 6870 freezer mill. Elongation inhibitors (CHX, ANS, and TIG) were used at 0.1 g/l. Lysed cell pellets were diluted to ~23 ml in yeast footprint lysis buffer containing specified antibiotics and clarified by centrifugation. The resultant supernatant was layered on a sucrose cushion [20 mM Tris–Cl (pH 8.0), 150 mM KCl, 5 mM MgCl$_2$, 0.5 mM DTT, 1M sucrose] to pellet polysomes in a Type 70Ti rotor (Beckman Coulter) (60,000 rpm for 106 min). Ribosome pellets were gently resuspended in 1 ml footprint lysis buffer (without elongation inhibitors). 400 µg of isolated

polysomes in 350 µl of footprint lysis buffer (without elongation inhibitors) were treated with 500 units of RNaseI (Ambion) for 1 h at 25°C. Monosomes were isolated by sucrose gradients (10–50%). The extracted RNA was size-selected from 15% denaturing PAGE gels, cutting between 15 and 34 nt. An oligonucleotide adapter was ligated to the 3′ end of isolated fragments. After ribosomal RNA depletion using RiboZero (Illumina), reverse transcription using SuperScript III reverse transcriptase (Thermo Fisher Scientific), circularization using CircLigase I (Lugicen), and PCR amplification. Libraries were sequenced on a HiSeq2500 machine at facilities at the Johns Hopkins Institute of Genetic Medicine.

### Polysome profiles

One litre of yeast cultures of indicated strains were harvested by fast filtration and ground with 1 ml footprint lysis buffer [20 mM Tris–Cl (pH 8.0), 140 mM KCl, 1.5 mM $MgCl_2$, 1% Triton X-100, 0.1 mg/ml CHX] in a Spex 6870 freezer mill. CHX was omitted for run-off polysome profiles. Cellular lysates were first clarified by centrifugation (at 15,000 rpm for 10 min). Lysates containing 200 µg of total RNA was spun through 10–50% sucrose gradients using a Beckman Coulter SW41 rotor at 40,000 rpm at 4°C for 2 h. Gradients were fractionated on a Biocomp piston gradient fractionator and the absorbance at 254 nm was recorded. Fractions were methanol precipitated and analyzed by immunoblotting using antibodies against eEF1 (Kerafast, ED7001), eEF2 (Kerafast, ED7002), and RPL4 (ProteinTech, 11302-1-AP).

### Tandem affinity purification of the eEF3-80S complex

The eEF3 *in vivo* pull-out was performed using DynabeadsR©M-270 Epoxy (Invitrogen) with yeast strain expressing a C-terminally TAP-tagged eEF3 (Strain: BY4741, Genotype: MATa *his3Δ1 leu2Δ0 met15Δ0 ura3Δ0*) obtained from Horizon Discovery. The purification was essentially performed as described previously (Schmidt *et al*, 2016). Briefly, cultures were harvested at log phase, lysed by glass bead disruption, and incubated with IgG-coupled magnetic beads with slow tilt rotation for 1 h at 4°C in buffer 30 (20 mM HEPES (pH 7.4). 100 mM KOAc, 10 mM Mg(OAc)$_2$, 1 mM DTT). The beads were harvested and washed three times using detergent containing buffer 30 (+0.05% Triton X) followed by a fourth washing step using buffer 30. The elution of the complex was done by addition of AcTEV Protease (Invitrogen) for 3 h at 17°C in buffer 30 containing 1 mM ADPNP (Sigma) final concentration.

### Sample and grid preparation

The final complex was cross-linked with 0.02% glutaraldehyde for 20 min on ice, and the reaction was subsequently quenched with 25 mM Tris–HCl pH 7.5. DDM (Sigma) was added to the sample to a final concentration of 0.01% (v/v). For grid preparation, 5 µl (8 $A_{260}$/ml) of the freshly prepared cross-linked complex was applied to 2 nm precoated Quantifoil R3/3 holey carbon supported grids and vitrified using a Vitrobot Mark IV (FEI, Netherlands).

### Cryo-electron microscopy and single-particle reconstruction

Data collection was performed on a FEI Titan Krios transmission electron microscope (TEM) (Thermo Fisher) equipped with a Falcon II direct electron detector (FEI). Data were collected at 300 kV with a total dose of 25 e$^-$/Å$^2$ fractionated over 10 frames with a pixel size of 1.084 Å/pixel and a target defocus range of −1.3 to −2.8 µm using the EPU software (Thermo Fisher). The raw movie frames were summed and corrected for drift and beam-induced motion at the micrograph level using MotionCor2 (Zheng *et al*, 2017). The resolution range of each micrograph and the contrast transfer function (CTF) were estimated with Gctf (Zhang, 2016). A total of 22,856 micrographs were collected. After manual inspection, 18,016 micrographs were used for automated particle picking with Gautomatch (http://www.mrc-lmb.cam.ac.uk/kzhang/) resulting in 530,517 initial particles, of which 211,727 were selected for further processing upon 2D classification in RELION-2.1 (Kimanius *et al*, 2016). After initial alignment with a vacant 80S reference, the 211,727 particles (defined as 100%) were 3D classified into 8 classes (Appendix Fig S3). Classes 1 and 2 (joined to group I) contained 70,780 particles (~33%) and displayed density for the eEF3-80S complex but had mixed tRNAs with varying occupancy as well as a dynamic L1-stalk. Class 3 (group II) showed a density for eEF2, whereas classes 4 and 5 (group III) revealed rotated ribosomal species with hybrid tRNAs. Both groups (group I and II) showed weak density for eEF3. Classes 6, 7 and 8 (group IV) (~31%) had low-resolution ribosomal species with biased orientation; however, all of them showed an extra density for eEF3 (not shown). To increase the resolution of the eEF3 ligand and separate it from low-resolution eEF3 species, group I was subjected to focused sorting using a mask encompassing the eEF3 ligand (Appendix Fig S3). The resulting class 3 containing 45,032 particles (~21%) was 3D and CTF refined using RELION-3.0 (Zivanov *et al*, 2018). The final refined volume was furthermore subjected to multi-body (MB) refinement, for which three masks were used: the first one encompassed one portion of eEF3 (ABC1, ABC2, and CD; residue range 420–976, MB-1) and the 60S, the second mask covered the 40S body (MB-2), and the third mask included the remaining part of eEF3 (HEAT and 4HB; residue range 1–419, MB3-3) as well as the 40S head (Appendix Figs S3 and S4F). The final reconstructions were corrected for the modulation transfer function of the Falcon 2 detector and sharpened by applying a negative B-factor estimated by RELION-3.0 (Zivanov *et al*, 2018). For the sharpening, a mask for the whole eEF3-80S complex was applied resulting in a final reconstruction of 3.3 Å (Appendix Figs S3 and S4A). The same was done for each part of the multi-body refined volumes, which provided a resolution of 3.2 Å for the 60S-eEF3 (ABC1/2, CD), 3.3 Å for the 40S body, and 3.5 Å for the 40S head-eEF3 (HEAT, 4HB) (Appendix Fig S4E). To obtain a stoichiometric tRNA occupancy as well as defined position of the L1-stalk interacting with the EF3-CD, group I was also subjected to further sorting into four classes using a flat cylinder mask encompassing the relevant regions (tRNAs, L1-stalk, and the eEF3-CD) (Appendix Fig S3). Each class was subsequently subjected to 3D and CTF refinement and a final postprocessing step. All the resulting classes were in an unrotated state bearing an A/A-, P/P- (Ia, 3.7 Å), P/P- (Ib, 3.8 Å), A/A-, P/E- (Ic, 4.0 Å), and a P/P- and E/E-site (Id, 3.9 Å) tRNAs. Particles of group II were extracted and subjected to CTF and 3D refinement resulting in a final cryo-EM reconstruction at 4.1 Å. For distinct tRNA occupancy of each 80S ribosome, group III (66,551 particles) was further 3D classified into two classes and each class was subsequently 3D and CTF refined. Both resulted in a ribosomal species with a disordered

eEF3 (IIIa, IIIb), which was visible after low pass filtering of the map. IIIa showed a rotated-1 state containing A/A- and P/E-site tRNAs with a final resolution of 4.2 Å. IIIb presented a classical fully rotated state (rotated-2) bearing a hybrid A/P- and P/E-site tRNA at 3.8 Å resolution. The resolutions for all volumes were estimated using the "gold standard" criterion (FSC = 0.143) (Scheres, 2012). Local resolution estimation and local filtering of the final volumes were done using Relion-3.0 (Appendix Fig S4B and C, F and G).

### Molecular modeling

The eEF3 model was based on the crystal structure of eEF3 in complex with ADP (PDB: 2IW3) with a 2.4 Å resolution (Andersen *et al*, 2006). The existing ribosome-bound eEF3 model (PDB: 2IX3) was used as a help for the rough fitting of the separate eEF3 domains (HEAT, 4 HB, ABC1, ABC2, and CD) of the crystal structure into the density (Andersen *et al*, 2006). The single domains were fitted with UCSF Chimera 1.12.1 (Pettersen *et al*, 2004) via the command "fit in map" and manually adjusted with Coot version 0.8.9.2 (Emsley & Cowtan, 2004). For the manual adjustment, the multi-body refined maps were used for the corresponding parts of the eEF3 model (MB-1 for ABC1, ABC2 and the CD; MB-3 for HEAT and 4HB). The model of the *S. cerevisiae* 80S ribosome was derived from PDB ID 6S47 (Kasari *et al*, 2019b) and the model for the L1 protein from the PDB ID 2NOQ (Schuler *et al*, 2006). The proteins of the 40S and 60S were fitted separately into locally filtered electron density maps using UCSF Chimera (Pettersen *et al*, 2004). The rRNA was fitted domain-wise in Coot (Emsley & Cowtan, 2004). Afterward, manual adjustments were applied to all fitted molecular models using Coot. The combined molecular model (proteins + rRNA) was refined using the phenix.real_space_refine command of phenix version 1.14 with restraints that were obtained via the phenix.secondary_structure_restraints command (Adams *et al*, 2010). Statistics for the model were obtained using MolProbity (Chen *et al*, 2010) and are represented in Appendix Table S1.

### Calculation of rotation angles and vectors

Rotation angles were calculated using UCSF Chimera with the command "match show matrix". The global rotation of the 18S rRNA was calculated relatively to the 23S rRNA by aligning all the models to the 23S rRNA of a non-rotated reference structure (PDB 6SNT) (Matsuo *et al*, 2020). The head swivel rotation degree was calculated relatively to the 18S rRNA body/platform by aligning all the model relative to the body of 18S rRNA from the reference structure (PDB 6SNT) (Matsuo *et al*, 2020). Vector calculation representing a shift between the phosphate atoms of the rRNA from the model compared with the reference structure was performed using PyMol Molecular Graphics System as previously described in (Beckert *et al*, 2018).

### Figure preparation

Figures showing biochemical experiments are fitted and plotted with GraphPad Prism 8.0.

Figures showing ribosome profiling data are created using R 3.3.1.

Figures showing atomic models and electron densities were generated using either UCSF Chimera (Pettersen *et al*, 2004) or Chimera X (Goddard *et al*, 2018) and assembled with Inkscape.

### Analysis of ribosome profiling data

Analysis of ribosome profiling data was performed as described previously (Wu *et al*, 2019a). The R64-1-1 S288C reference genome assembly (SacCer3) from the *Saccharomyces* Genome Database Project was used for yeast genome alignment. Ce10 reference genome assembly from UCSC was used for *C. elegans* genome alignment. Hg19 reference genome assembly from UCSC was used for human genome alignment. A human transcriptome file was generated to include canonical transcripts of known genes from UCSC genome browser. WT replicate 1 (CHX + TIG) and both WT replicates (CHX + ANS) are identical to that published previously (Wu *et al*, 2019b). Libraries were trimmed to remove ligated 3′ adapter (NNNNNNCACTCGGGCACCAAGGA), and 4 random nucleotides included in RT primer (RNNNAGATCGGAAGAGCGTCGTGTAGG GAAAGAGTGTAGATCTCGGT. GGTCGC/iSP18/TTCAGACGTGTGCT CTTCCGATCTGTCCTTGGTGCCCGAGTG) were removed from the 5′ end of reads. Trimmed reads longer than 15 nt were aligned to yeast ribosomal and non-coding RNA sequences using STAR (Dobin *et al*, 2013) with "--outFilterMismatchNoverLmax 0.3". Unmapped reads were then mapped to genome using the following options "--outFilterIntronMotifs RemoveNoncanonicalUnannotated --outFilterMultimapNmax 1 --outFilterMismatchNoverLmax 0.1". All other analyses were performed using software custom written in Python 2.7 and R 3.3.1.

For each dataset, the offset of the A site from the 5′ end of reads was calibrated using start codons of CDS (Schuller *et al*, 2017). Relative ribosome occupancies (pause scores) for codons or peptide motifs (Fig 3E–H) were computed by taking the ratio of the ribosome density in a 3-nt window at the codon or motif of interest over the overall density in the coding sequence (excluding the first and the last 15 nt to remove start and stop codons). Peptide motif logos (Appendix Fig S3E) were generated with WebLogo (Crooks *et al*, 2004) by using motifs with a pause score greater than 4.

### Cryo-EM data analysis

Bayesian selection using RELION software package was used to choose the cryo-EM data package (Scheres, 2012). Resolutions were calculated according to gold standard, and the estimation of variation within each group of data was performed using Bayesian calculation within RELION (Scheres, 2012).

## Data availability

Cryo-EM maps generated during this study have been deposited in the Electron Microscopy Data Bank (EMDB; https://wwwdev.ebi.ac.uk/pdbe/emdb) with accession codes EMD-12081 (eEF3-80S complex), EMD-12059 (PRE-1), EMD-12061 (PRE-2), EMD-12075 (PRE-3) and EMD-12065 (PRE-4), EMD-12074 (POST-1) and EMD-12062 (POST-2), EMD-12064 (POST-3). The eRF3-80S complex model generated during this study has been deposited in the Protein Data Bank (PDB; http://www.wwpdb.org) with accession code

7B7D. Raw data for ribosome profiling have been deposited to the Gene Expression Omnibus (https://www.ncbi.nlm.nih.gov/geo) with the accession number GSE160206.

**Expanded View** for this article is available online.

## Acknowledgements

We thank Prof. Ralf Ficner for providing the strains for eIF1 and eIF1A purification. We thank Theresia Steiger, Tessa Hübner, Olaf Geintzer, Susanne Rieder for expert technical assistance, Paul Huter for help with data processing, and Otto Berninghausen and Roland Beckmann (Gene Center, LMU, Munich) for data collection. This work has been supported by iNEXT (project number 2643 to D.N.W.), the Horizon 2020 program of the European Union (CEITEC MU), NIH (R37GM059425 to R.G.), HHMI (R.G and C.C.W.), and grants of the Deutsche Forschungsgemeinschaft (DFG) WI3285/8-1 to D.N.W., RA 3194/1-1 to N.R., and Leibniz Prize to M.V.R.. This article reflects only the author's view and the European Commission is not responsible for any use that may be made of the information it contains. CIISB research infrastructure project LM2015043 funded by MEYS CR is gratefully acknowledged for the financial support of the measurements at the CF Cryo-electron Microscopy and Tomography CEITEC MU. Open Access funding enabled and organized by ProjektDEAL.

## Author contributions

Biochemical and kinetic experiments: NR, SB; Ribosome profiling experiments and analysis: CCW; Cryo-EM analysis: AAP, BB; Manuscript writing: NR, SB, AAP, CC-CW, RG, MVR, DNW.

## Conflict of interest

The authors declare that they have no conflict of interest.

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
