## [Review Process File · The EMBO Journal]

Yeast translation elongation factor eEF3 promotes late stages of tRNA translocation

Namit Ranjan, Agnieszka Pochopien, Colin Wu, Bertrand Beckert, Sandra Blanchet, Rachel Green, Marina Rodnina, and Daniel Wilson

DOI: [10.15252/embj.2020106449](https://doi.org/10.15252/embj.2020106449)

Corresponding author(s): Daniel Wilson (Daniel.Wilson@chemie.uni-hamburg.de), Rachel Green (ragreen@jhmi.edu), Marina Rodnina (rodnina@mpibpc.mpg.de), Namit Ranjan (namit.ranjan@mpibpc.mpg.de)

Review Timeline:

Submission Date:	6th Aug 20
Editorial Decision:	10th Sep 20
Revision Received:	3rd Nov 20
Editorial Decision:	1st Dec 20
Revision Received:	10th Dec 20
Accepted:	21st Dec 20

Editor: Stefanie Boehm

Transaction Report:

Thank you for submitting your manuscript for consideration by The EMBO Journal. Please also excuse the delay in communicating the decision to you, which was due to delayed referee reports on account of the current pandemic as well as absences over the summer holiday period. We have now however received three referee reports on your study, which are included below for your information.

As you will see, the reviewers are overall positive and acknowledge the interest of the field in the topic and the study. Nonetheless they also raise several concerns that should be addressed in a revised manuscript. In particular, additional controls and discussion are required for Figure 2, where both referee #1 and #2 are not fully convinced by the conclusion that dissociation of the E-site tRNA is not coupled to TC binding and indicate potential alternative explanations, which must be addressed. In addition, all referees note several instances where additional clarifications regarding data analysis, statistics and replicates are needed (ref #1- points 1, minor 1, 2, 3, 4; ref #2 specific comments Fig. 2, 3; ref #3 point 2, minor points). Moreover, referee #3 raises questions regarding the role of ATP hydrolysis and the use of AMP-PNP (ref #3- points 1, 3), which should be addressed, also taking into account referee #1's suggestion 1. Please also carefully consider all remaining other points of the referees and provide additional data or further discussion and clarification in the revised version. If you are able to adequately address these concerns we would be happy to consider the manuscript further for publication. Therefore, I would now like to invite you to prepare and submit a revised manuscript.

REFEREE REPORTS

Referee #1:

The study by Ranjan et al. contains a beautiful combination of biochemical, ribosome profiling and structural data to investigate the function of the eukaryotic translation elongation factor eEF3 in fungi. The biochemical and ribosome profiling data show that eEF3 promotes efficient tRNA translocation during elongation by facilitating ejection of the de-acylated E site tRNA. In addition, the authors present multiple structures of ribosome-bound eEF3 using single-particle cryo-EM. The structural data indicate that eEF3 binding depends on the rotational state of the ribosome and may control ejection of the E site tRNA by influencing the conformation of the ribosomal L1 stalk. Overall, the experimental approaches are of high quality and the study provides valuable and novel insights to foster our understanding of translation regulation in eukaryotes.

Prior to publication, the authors should however address the following:

Major concerns:

1. Fig.1 B-C and Fig.2 B: The scale of the Y axis is not clear. The authors have probably normalized their data but it is not defined in the figure legend or the Material and Methods section how they have done so. It therefore is also not clear what percentage of the isolated 80S IC is actually active in the kinetic experiments, which should be included in the manuscript to validate their findings.
2. From the experiment shown in Fig.2 D, the authors draw the conclusion that TC binding and E site tRNA release are not coupled. However, there is no control experiment included that shows how much of the E site tRNA is still bound at the start of the kinetic measurement and how much may already have dissociated due to the long incubation time of 15 min before addition of the TC. This is a crucial information that has to be added to judge the validity of this conclusion. In case most of the E site tRNA is gone already, such a conclusion could not be drawn from this experiment.
3. The model vs map FSC should be included in Appendix Fig. S5 to indicate the fit of the structural model to the experimental data over all resolution shells.

Minor concerns:

1. Fig. 1D: The brown triangles are not labeled in the figure legend and the identity of the sample remains unclear to the reader. The authors should include the appropriate labelling.
2. Fig. 2C and D: The authors could add a scale to the Y axis.
3. In POST-1 the tRNA states are denoted as P/P and E/E. However, with the head of the SSU swiveled one would expect rather chimeric ap/P and pe/E states. If this is the case, the authors should correct their nomenclature.
4. Please clarify which CC is given in Table S1: CCmask?, CCbox? Or other?

Suggestions:

1. The biochemical data show convincing evidence for a role of eEF3 in controlling E site tRNA ejection. Interestingly, the authors discover a contact between the L1 stalk and eEF3 that appears to be based on charge complementarity. It is not so clear how this attracting interaction is used by eEF3 to facilitate opening of the L1 stalk and tRNA release as the authors propose. Perhaps eEF3 rather fosters an inward-facing, closed conformation of the stalk. The mechanistic model may be improved if the authors could include a speculation about the mechanism of stalk opening. Does ATP hydrolysis play a role, or ribosomal rotation?
2. The following sentence is confusing: "Analogous to PRE-1 and PRE-2, both PRE-3 and PRE-4

states also had density for the nascent polypeptide chain, but extending from the A site (rather than the P-site tRNA) into the ribosomal exit tunnel." It may be misinterpreted as the PRE-1 and PRE-2 states contain the nascent chain being bound to the P site tRNA, which is not the case. The authors should clarify this sentence.

Referee #2:

Previous studies have suggested that the fungal-specific elongation factor EF-3 acts to stimulate coupled release of deacylated tRNA from the E site and binding of ternary complex (TC) to the A site; and structural analysis of an EF-3/80S complex revealed the EF-3 interaction surfaces on the back of the 40S and 60S subunits. This paper presents a combination of biochemical analysis of elongation in a purified system, ribosome profiling in EF-3 depleted cells and cryoEM analysis of various native EF-3 bound 80S complexes in an effort to better establish the mechanism of EF-3 function in the elongation cycle.

By monitoring rates of di- or tri-peptide synthesis in a fully reconstituted system, the results in Fig. 1C-D show that EF-3 has no effect on the rate of dipeptide synthesis, but it greatly accelerates tripeptide synthesis. The results in Fig. 1D suggest that EF-3 does not stimulate the rate of TC binding to the A site following EF-2-catalyzed translocation of the dipeptidyl-tRNA to the P site. By following the rate of puromycin incorporation, the results in Fig. 2A suggest that EF-3 has relatively little effect on the rate of peptide bond formation. By directly monitoring the rate of dissociation of labeled E-site tRNA, the findings in Fig. 2C indicate that EF-3 promotes dissociation of E-site tRNA following EF-2 catalyzed translocation of dipeptidyl-tRNA, even when binding of TC to the A-site is not occurring (purple curve in Fig. 2C). The experiments in Fig. 2D indicate that binding of TC alone following EF-2 catalyzed translocation is insufficient to evict the E site tRNA, but which occurs efficiently when EF-3 is also present. The authors conclude that EF-3 promotes release of E-site tRNA but has no role in stimulating TC binding to the A site, and that these two events are not coupled in the manner concluded from previously published experiments. Overall, these findings suggest that EF-3 stimulates the rate of elongation by 10-fold or more (depending on the codons involved) by promoting dissociation of E-site tRNA in a late stage of translocation.

Ribosome profiling results in cells depleted of EF-3 revealed a depletion of elongating ribosomes with an empty A site (with 21nt RPFs) at the expense of those with occupied A sites (with 28nt RPFs) and the use of different antibiotics in the extracts led to the conclusion that the pre-translocation complexes following peptide bond formation dominate the 28 nt RPFs, consistent with a rate-limiting defect in translocation-which is in agreement with the biochemical data. Examining codon pause scores indicated a loss of pauses at non-optimal codons in the A site in the 21 nt RPF data; and a loss of pausing at Pro, Gly, and Asp codons in the P site in the 28 nt RPF data, upon depletion of EF-3. These results were attributed to translocation becoming rate limiting rather than A site decoding for poor codons, and presumably rather than a different step in translocation that is normally slow at P,G and D codons, upon EF-3 depletion.

The third part of the paper describes cryoEM analysis of 80S ribosomes affinity purified from cells using TAP-tagged EF-3. A variety of different states were observed, some containing well-resolved EF-3 and others not. Analyzing the former shows EF-3 binding to the 40S and 60S subunits in a location and manner expected from the previous structural analysis of EF-3/80S complexes by Anderson et al. Among these, two represent post-translocation states based on the presence of peptidyl tRNA in the P site, but only one of the two contains an E site tRNA. Comparing these last two suggests that dissociation of E site tRNA is accompanied by movement of 60S protein L1 to an "out" configuration in which it is no longer engaged with the E tRNA. Two other complexes with well-resolved EF-3 contain A and P site tRNAs with the nascent peptide on the A site tRNA,

signifying pre-translocation complexes, suggesting the EF-3 binds throughout the elongation cycle, even though it seems to affect only the late stage of translocation by helping to release the E site tRNA. Three other complexes containing poorly resolved EF-3 or no EF-3 density at all were also analyzed, two in pre- and one in post-translocation configurations based on the tRNAs present in the decoding sites. Interestingly, all of the latter feature intersubunit rotation, which is thought to occur during translocation; whereas the four complexes containing well-resolved EF-3 all represent non-rotated states. These findings suggest that stable EF-3 binding requires the non-rotated state of the ribosome, which is consistent with loss of EF-3 contacts with the 40S head predicted for the rotated 80S ribosome. Because two of the non-rotated, EF-3-bound complexes appear to be post-translocation complexes based on containing peptidyl-tRNA in the P site, the authors propose that EF-3 preferentially binds to, or stabilizes 80S ribosomes that have switched from rotated back to non-rotated subunit interfaces following translocation, and that EF-3 then catalyzes dissociation of E site tRNA from this non-rotated state by displacing L1 to its "out" conformation. This model envisions that EF-3 remains tightly bound to the non-rotated ribosomes through TC binding to the A site and peptide bond formation, binds weakly as intersubunit rotation proceeds and during EF-2 catalyzed translocation, and then resumes stable binding when the ribosome switches back to the unrotated state (or promotes this switch), where it catalyzes release of the E-site tRNA. The biochemical data, and ribosome-profiling data imply that EF-3 only significantly stimulates E-site tRNA release and does not accelerate TC recruitment to the A site or peptide bond formation even though it appears to be stably bound to the ribosome at all of these steps.

General Critique: There is a wealth of valuable data in this report obtained by a very interdisciplinary approach that draws on different strengths of the three labs involved, and which appear to significantly advance our understanding of EF-3 function. The data and final model depart substantially from the previous conclusion that EF-3 catalyzes a coupled release of E site tRNA and A site binding of TC with evidence that these events do not exhibit obligatory coupling and that EF-3 can stimulate tRNA dissociation from the E site in the absence of TC, and no apparent acceleration of TC binding to the A site given by EF-3. The ribosome profiling data support a defect in translocation that would be expected from defective E-site tRNA dissociation, and the cryoEM structures provide evidence consistent with EF-3 catalyzing dissociation of the E-site tRNA without the presence of TC in the A site.

The manuscript is difficult to read however and the presentation could be considerably improved to explain things better, improve the figures, and eliminate incorrect or confusing statements in all three sections of the paper. In addition, while the profiling data provide evidence that EF-3 depletion alters the rate-limiting steps in elongation, evidence is lacking that EF-3 depletion actually reduces the rate of translation at the elongation stage *in vivo*. For some of the analyses of ribosome profiling data, it is unclear whether biological replicates provide consistent results and support the derived conclusions. And there are deficiencies in the analysis of bulk polysomes and polysome association of EF-1A and EF-2 upon EF-3 depletion from cells. There are also questions about whether appending the tag to EF-3 alters its function *in vivo*, and whether the complexes analyzed by cryoEM were all truly enriched for EF-3, as an untagged control strain was apparently not employed in the purification experiments.

Specific comments:

-The description of the kinetic data in Figs. 1-2 is rather dense, and would benefit from a more expansive treatment indicating more clearly the specific question being addressed by each experiment regarding whether a particular step in elongation is stimulated by EF-3, and the features of the assay that allow that step to be studied in isolation of the others, followed by a succinct conclusion reached from the findings before moving on to the next partial reaction under consideration. The next five comments are all related to this one.

-Fig. 2A could be improved and a detailed explanatory legend provided that better matches the narrative in the text. It does not do a good job of depicting the different stages of translocation as described in the text on the top of p. 7. CCW and CW are not defined in the legend, nor are PRE and POST; and the hybrid and "chimeric" states are not indicated in the figure. The three steps described in the text are not mentioned explicitly in the figure. Whether the tRNAs are in the P/P or P/E states could also be shown.

-It would be useful to add a sentence towards the end of p. 7 giving the conclusion about the data in Fig. 2B regarding the importance of EF3 for the rate of translocation, ie. that EF3 is dispensable for efficient translocation but enhances the rate of the reaction.

-It might be helpful to add to Fig. 2A a schematic of a reaction depicting E site clearance in the absence of TC-Val, which EF-2 and EF-3 can catalyze based on the purple curve in Fig. 2C.

-p. 9: the sentence "We further tested whether the tRNA dissociation from the E site and TC binding to the A site are directly coupled." might be improved by writing "We further tested whether the tRNA dissociation from the E site and TC binding to the A site are directly coupled in the absence of EF-3". The sentence "These experiments clearly establish that binding of the cognate TC into the A site is not coupled to the dissociation of the E-site tRNA" might be improved by writing "These experiments clearly establish that binding of the cognate TC into the A site alone following EF-2 catalyzed translocation, is not sufficient to promote dissociation of the E-site tRNA"

-The authors conclude that EF-3-catalyzed dissociation of E-site tRNA is not coupled to TC binding to the A site; however, is it possible that the more rapid and complete dissociation of E-site tRNA observed in the presence versus absence of TC shown in Fig. 2C (green vs purple curves) is an indication that coupling of the two reactions increases the efficiency of EF-3 function?

-p. 9: indicate that the inhibitors used in ribosome profiling are added to the cell extracts to prevent interconversion between the states shown in Fig. 3A following cell lysis and thereby help to preserve the distribution among the states present in vivo.

- p. 10: For the sentence "By contrast, in eEF3d cells, we observed a drastic reduction in 21 nt RPFs (and a concomitant increase in 28 nt RPFs) when compared to WT cells (Figures 3B and Appendix Figure S3C), indicating the accumulation of PreTrans state ribosomes." Isn't it more correct to say that based on these data there is an accumulation of ribosomes in either the PrePT or PreTrans states? The next experiments with anisomycin are used to distinguish between these two states by blocking conversion of PrePT to PreTrans states in the extract, which allows accurate quantification of the PrePT state in vivo. If my interpretation of this is correct, it would be helpful to introduce the anisomycin experiments in Figs. 3C-D in that way.

-p.10: the sentence "We further found that the footprints from eEF3d cells are almost devoid of 21 nt RPFs when compared with the WT samples prepared with CHX+ANS (Figure 3D)." should be followed by an interpretative comment to the effect that because 28nt RPFs originate solely from PreTrans 80S complexes owing to dissociation of PrePT 80S ribosomes when ANS is present, these results indicate an overabundance of PreTrans complexes on EF-3 depletion from cells, consistent with impaired translocation rather than a defect in peptide bond formation in vivo.

-Fig. 3A-D and associated text: It's unclear whether the results shown here were obtained reproducibly in biological replicates. Ideally, the mean proportions of RPFs of different fragment lengths should be presented with errors and statistical analysis from replicate experiments. At least, curves like those in panels B and D should be shown for each of the replicates.

-Fig. S3D: It should be stipulated whether CHX was added to cells to freeze elongation in these experiments. (In fact, there are no Methods given for this experiment or the others in Fig. S3.) Also, the Western signals for EF-1A and EF2 on polysomes are lower in the mutant vs. WT. To conclude, as they do, that association of EF-1A and EF2 with polysomes is normal on EF-3 depletion, the western signals would have to be normalized to the A260 tracings, or to Western signals for a ribosomal protein, and results presented from replicate experiments. It would also be reassuring to see that the polysomal signals for these factors is eliminated on RNase treatment to collapse

polysomes to monosomes to rule out non-polysomal aggregates contributing to the polysome signals.

-Assuming CHX was added to cells for the experiment in Fig. S3D, it is surprising that the polysomes are depleted in the mutant rather than accumulating and showing a shift to larger polysomes in the manner expected for reduced rates of elongation. Rather, polysomes relative to monosomes are depleted, indicating reduced initiation, or wholesale mRNA degradation, which could occur as a secondary response to 8 h of EF-3 depletion. It seems important to compare polysome profiles in the presence and absence of CHX, as a reduced rate of elongation in the mutant should diminish polysome run-off in the absence of CHX compared to what occurs in WT cells, which would provide direct evidence for reduced elongation rates in vivo on depletion of EF-3.

-p. 11 bottom and Figs. 3G-H: It's difficult to comprehend the loss of P site pausing at P, G, and D codons as determined by analysis of 28nt RPFs in the experiments of Fig. 3G and H. Since CHX + ANS enriches for PreTrans 80Ss, the implication appears to be that translocation, rather than peptide bond formation, is slow with these P site codons in WT cells. But if EF-3 is slowing down translocation, one might have expected exacerbation of the slow translocation rates at these codons. Do the authors have to postulate that a different step in translocation is impaired which becomes more rate-limiting on EF-3 depletion distinct from the aspect of translation that is slow in WT cells at these codons? A more illuminating interpretation of these results is needed.

-the legend to Fig. S3E doesn't stipulate whether A or P site pauses are being represented, and other details about how these motifs were determined are lacking; and if they were observed in both replicates.

-Fig. 4A doesn't stipulate what antibody is used in the Western blot. One would like to see that the supernatant lane would be essentially blank from a parallel pull-down of the WT untagged strain to be sure that the recovery of this band and associated RPs is dependent on the EF-3 pull-down. This seems important since the presence of EF-3 is either questionable or lacking in the complexes shown in Fig. 6A, C, D but it is assumed that they contained EF-3 during the pull-down.

-Figs. S7E-G don't do a good job of showing the predicted clash of the EF-3 domain with the ribosome for a predicted open conformation. A blow-up of the predicted clashing elements seems necessary. Also the predicted dissociation seems to be restricted to the cyan HEAT domain, the only component of the ABC1-HEAT-4HB domain that contacts the 40S, which would be helpful to point out more explicitly.

-p.11: "...the N-terminal HEAT and 4HB interact with the 40S subunit, whereas the C-terminal CD contacts exclusively the 60S subunit." This sentence is misleading in implying 4HB interactions with the 40S.

-p.13-14: Indicate whether the specific contacts of EF3 HEAT, CD, and ABC2 domains with the ribosome mentioned here were observed previously in Anderson et al, or instead newly observed here.

-Analysis needs to be provided to show whether appending the TAP tag to EF-3 affects cell growth or polysome profiles.

-p. 18: I take issue with the sentence "Similar to previous studies where eEF3 was depleted in the cell (Kasari et al., 2019a), we observe general defects in translation elongation." as the polysome profiling shown in Fig. S3 reveal an initiation defect and they have not provided evidence that the rate of elongation is reduced in vivo on EF-3 depletion. The ribosome profiling data indicate that EF-3 depletion appears to be altering the rate-limiting steps in the elongation cycle, but they don't indicate directly that the rate of translocation is reduced from its WT rate and that this reduces the overall elongation rate. As such, using the word "trapped" in the following sentence is not justified.

-p. 18, regarding the sentence: "A strong prediction of the biochemical and ribosome profiling results is that the ribosomes that accumulate in cells on eEF3 depletion would contain three separate tRNAs." shouldn't Fig. 7H be cited here showing this aberrant species? Also, shouldn't this aberrant state be generated from the pre-translocation intermediate in panel D vs. the post- intermediate in

panel E with EF-2 in the A site? Later on p. 19, the sentence " we envisage that ribosomes are trapped in the POST-2 state with deacylated tRNAs in the E- and P-sites and a peptidyl-tRNA in the A site." seems to indicate that the state in Panel H would originate from that in Panel F. Either the thinking or the writing on this point seems to be unclear.

-Fig. 7 is confusing in that two arrows are not shown between panels G and A representing TC binding to A site (by yellow tRNA shown) and peptide bond formation (arrow missing) that switches the peptidyl-tRNA from the P site (red) to A site (yellow). Also wording at the end of the legend is awkward and confusing.

-p. 18-19: in the phrase "whereas dissociation of the factor and back swivel occur in a later step (Figure 7A-B)." shouldn't Fig. 7F-G be cited instead of A-B?

Referee #3:

Review of EMBOJ-2020-106449

Ranjan et al. address the yet vaguely defined function of the translation elongation factor eEF3, which is essential in fungi but absent in most other eukaryotes. Using a fully reconstituted translation elongation system from the bona fide model organism *Saccharomyces cerevisiae*, Ranjan et al. show that eEF3 is responsible for E-site tRNA ejection independent of other translation elongation factors. Ribosome profiling in eEF3-depleted bakers' yeast suggests a general role for eEF3 in translation elongation. Ex vivo cryo-EM structures at 3-4 Å resolution partially complement the known translation elongation cycle structurally and allow insights into the molecular mechanism of eEF3. It is important to note that the X-ray crystal structure of yeast eEF3 (Andersen et al. 2006) has been used to fit the cryo-EM map, which complements a study on New1p (Kasari et al. 2019 NAR). Unfortunately, the authors could only visualize eEF3 at 80S in a single conformation after glutaraldehyde crosslinking and trapping by non-hydrolysable AMPPNP, so both the chemomechanical coupling and the function of the two nucleotide-binding sites of eEF3 remain unknown.

The manuscript by Ranjan et al. is generally well written and very interesting for the scientific community. The data are experimentally diverse, supplement each other and are presented in a comprehensible way. However, some concerns need to be addressed to support the overall conclusions of the authors.

Major issues:

1) Biochemical in vitro studies: The authors do not state that (or how much) ATP was used in the in vitro experiments. This point is critical because the ATP-occluded state has been described as ribosome-bound in this and previous studies. In fact, if ATP was not present, eEF3 might have used GTP from the TC preparation. In this case, the authors should show in additional experiments (e.g. like Fig. 1C, D and 2C) that ATP does not alter the outcome. In addition, they could provide ATPase and GTPase assays (colorimetric or radiographic) to show that eEF3 can utilize GTP in a similar way as ATP, which might well be true for an ABC-type system. Since eEF3 has been used in excess, even a low, unspecific: GTPase activity could suffice to accelerate the reaction, but the kinetic values would not reflect the physiological situation, which should be clearly stated. If control experiments with ATP have to be repeated, I suggest that the authors additionally use ADP and AMPPNP to correlate the function of eEF3 in the translation cycle with its ATPase activity, similar to

what has already been done for Rli1/ABCE1, for example.

2) Ribosome profiling studies: The authors should exclude that the levels of eIF5A have dropped significantly upon eEF3 depletion. Given a half-life of 9.1 h (source: <https://yeastmine.yeastgenome.org>) and impaired translation, the effects seen in the ribosome profiling experiments after 8 h may reflect partial loss of eIF5A. Especially, since the codon-resolved analysis showed an overlap with eIF5A function. The authors could easily verify this by using the samples from Fig. S3A for an immunoblot against eIF5A.

3) Structural data - Page 17-18 and Fig. 7:

„This is somewhat surprising since formation of hybrid states in higher eukaryotes is generally assumed to be concomitant with intersubunit rotation (Behrmann et al., 2015; Budkevich et al., 2011; Svidritskiy et al., 2014), however, we note that subunit rotation and tRNA movement is only loosely coupled on bacterial ribosomes (Fischer et al., 2010). Since rotation of PRE- 2 would result in a state similar to PRE-3, we suggest that subunit rotation is what triggers the low affinity form of eEF3, effectively leading to unstable ribosome binding."

„The ability to capture four different PRE and three different POST states of the ribosome by co-immunoprecipitation of tagged-eEF3 suggests that eEF3 is omnipresent during translation elongation in yeast..."

„(C) Rotation of the ribosomal subunits leads to unstable binding of eEF3 to a ribosome with A/A- and P/E-tRNA (PRE-3) as well as to (D) a fully rotated ribosomal species bearing hybrid A/P- and P/E-tRNAs (PRE-4)"

The authors should discuss the point that usage of AMP-PNP and a chemical crosslinker might stabilize non-physiological states of the 80S-eEF3 complex or even complexes that are never formed in vivo. This is reminiscent of the situation with ABCE1 in translation initiation complexes, which in structural studies is always bound to 40S subunits in the presence of AMP-PNP (see e.g. Simonetti et al., 2020 or Heuer et al., 2017), but dissociates upon ATP addition (see e.g. Simonetti et al., 2020). A recent BioRxiv preprint (Kratzat et al., 2020) now demonstrates ABCE1 in initiation complexes in a new state, possibly manifesting its role in translation initiation, but certainly showing that AMP-PNP-containing samples cannot be used to extend structural data to functional knowledge without validation by biochemical experiments. If this cannot be ruled out by convincing arguments or references, the authors should relativize the respective statements. Alternatively, cryo-EM experiments (low resolution should be sufficient) could be performed using ex-vivo samples without AMP-PNP. The authors could also consider biochemical (e.g. co-immunoprecipitation) or biophysical (e.g. fluorescence-based) eEF3-binding studies of defined 80S complexes (in the presence of antibiotics) to support their statements.

Minor points

1) Fig. S1A (left panel): The authors should state what "nat" means.

2) Fig. S2A: Please indicate how much 80S IC was applied to directly correlate the 2 μ M and 4 μ M eEF3 concentration.

3) Fig. 1D: What are the brown triangles? Puromycin test as in Fig.2?

4) Page 6 and Fig. 1D „(ii) eEF2 alone can complete the translocation process given enough time" :

The Met-Phe-Val formation is not complete with eEF2 alone, which the authors also state on page 9: „eEF2 alone promotes partial tRNA translocation" : In Fig. 1D some data points are missing at the end of the red trace. The authors should comment on why they were removed or not analyzed in this experiment. The authors should clarify their message and state that translocation by eEF2 alone is either slow but complete and the difference is within the range of error or it is slow and incomplete.

5) Page 7 and Fig. 2B: „We note that in the presence of eEF3 and eEF2, the reaction with Pmn was faster than with eEF2 alone, suggesting that binding of eEF3 to the ribosome stabilizes a ribosome conformation that is somewhat more active in peptidyl transfer reaction with Pmn, possibly by stabilizing peptidyl-tRNA in the exit tunnel": However, the reaction is incomplete (app. 75%) in the presence of both factors eEF2 and eEF3 (green) compared to eEF2 alone (red). Thus, eEF3 seems to have an inhibitory role on a portion of ribosomes during the Pmn reaction, which should be mentioned here. Interestingly, the end point defect is comparable to the one in Figure 1D, which may indicate heterogeneity in the 80S 2C population.

6) Fig. S2G: The labeling is a bit irritating. It looks like only eEF3 was added because in all previous figures, all the components were listed. I suggest removing eEF3 here, leaving just the tRNA labels or adding TC and eEF2 to be consistent with the other figures.

7) Page 8 and Fig. 2C: „When a similar experiment is carried out in the absence of TC-Val, only the first translocation can occur; we observe that in this case, the fluorescence of tRNA^{Met} decreases considerably, but not to the same extent as when TC-Val is included (Figure 2C, purple trace)": The authors should consider 80S 2C heterogeneity as a reason for the difference in E-site tRNA clearance in the absence and presence of the TC. If ATP was not present in the reaction mixture, the absence of GTP could be the reason for the incomplete reaction of the purple trace.

8) Page 9: The study by Velechano & Alepuz (NAR, 2017) should be mentioned in the context of the global role of eIF5A in translation.

9) Page 13 "archaeal 40S-ABCE1 complex (Heuer et al., 2017)":

The archaeal complex with the highest resolution (2.8 Å) obtained for SSU-ABCE1 (which is a 30S ribosomal subunit in Archaea, not a 40S) was described in Nürnberg-Goloub et al., 2020, EMBO J. Heuer et al. 2017 NSMB describes the yeast 40S post-splitting complex. The author might consider to include an additional reference for the open conformation of ABCE1 (Barthelme et al. 2011 PNAS).

10) Page 13: "...ribosome-stimulated hydrolysis of ATP to ADP would promote eEF3 dissociation": The authors should provide a reference, which describes the stimulation of eEF3 ATPase activity by the ribosome and, putatively, its dissociation from the ribosome upon ATP hydrolysis. Alternatively, they could provide examples from other ABC-proteins (ABCF? ABCE?) showing such behavior to support their statement.

11) Fig. S6D: The authors clearly present the presence of a nucleotide, which is most likely AMPPNP based on sample preparation and overall conformation of eEF3. However, ABC proteins essentially occupy Mg²⁺ ions with each nucleotide. The authors should show the density for the ions in this figure or comment on why the density is absent from their sample.

Referee #1:

The study by Ranjan et al. contains a beautiful combination of biochemical, ribosome profiling and structural data to investigate the function of the eukaryotic translation elongation factor eEF3 in fungi. The biochemical and ribosome profiling data show that eEF3 promotes efficient tRNA translocation during elongation by facilitating ejection of the de-acylated E site tRNA. In addition, the authors present multiple structures of ribosome-bound eEF3 using single-particle cryo-EM. The structural data indicate that eEF3 binding depends on the rotational state of the ribosome and may control ejection of the E site tRNA by influencing the conformation of the ribosomal L1 stalk. Overall, the experimental approaches are of high quality and the study provides valuable and novel insights to foster our understanding of translation regulation in eukaryotes. Prior to publication, the authors should however address the following:

Major concerns:

1. Fig.1 B-C and Fig.2 B: The scale of the Y axis is not clear. The authors have probably normalized their data but it is not defined in the figure legend or the Material and Methods section how they have done so. It therefore is also not clear what percentage of the isolated 80S IC is actually active in the kinetic experiments, which should be included in the manuscript to validate their findings.

We provided the information on the way we normalize the data in legends to Figs. 1 and 2 and Fig. EV1 and included the fraction of purified 80S IC that was active in the kinetic experiments in Materials and Methods, p. 38.

2. From the experiment shown in Fig.2 D, the authors draw the conclusion that TC binding and E site tRNA release are not coupled. However, there is no control experiment included that shows how much of the E site tRNA is still bound at the start of the kinetic measurement and how much may already have dissociated due to the long incubation time of 15 min before addition of the TC. This is a crucial information that has to be added to judge the validity of this conclusion. In case most of the E site tRNA is gone already, such a conclusion could not be drawn from this experiment.

We thank the reviewer for a very good suggestion. We performed the control experiments suggested by the reviewer to monitor the amount of the E-site tRNA that remains bound at the start of the kinetic experiment after 15 min of incubation with eEF2 alone or eEF2/eEF3. We incubated 80S 2C (containing fluorescence labeled tRNA^{fMet} in the P site and Met-Phe-tRNA^{Phe} in the A site) in the absence of eEF2 or eEF3 (control for stable tRNA^{fMet} binding), or after incubation with eEF2 alone or with eEF2+eEF3 for 15 min in the absence of TC-Val and monitored the amount of the E-site tRNA retained on the 80S 2C by tracking the fluorescence of tRNA^{fMet}(Flu) in size exclusion chromatography. The sample with eEF2, eEF3, and TC-Val served as a control for residual tRNA^{fMet}(Flu) binding after two rounds of translocation. We show that after incubation for 15 min with eEF2 alone, only small fraction of the E-site tRNA dissociates, whereas incubation with both eEF2 and eEF3, which allows one round of translocation to happen, results in a significant dissociation of the E-site tRNA during the gel filtration. The amount of fluorescence tRNA co-eluting with the ribosome is reduced further upon addition of TC-Val due to Val-tRNA binding to the A site, tripeptide formation, and the second round of translocation. These data are now included in Fig 2D. Former Fig. 2D (now 2E) provides the kinetic evidence for the lack of E-site tRNA dissociation with eEF2 alone when TC-Val is added. On the other hand, Fig. 1D shows that incubation for 15 min with eEF2 alone and addition of TC-Val result in tripeptide formation,

indicating that Val-tRNA^{Val} can bind and take part in peptide bond formation even if the E-site tRNA is not released (Fig. 2D,E). These results support our conclusion that TC binding and E-site tRNA release are not coupled. The additional text was added on p. 7-8.

3. The model vs map FSC should be included in Appendix Fig. S5 to indicate the fit of the structural model to the experimental data over all resolution shells.

We have now provided model vs map FSC graphs as requested as new panel A in Appendix Figure S4.

Minor concerns:

1. Fig. 1D: The brown triangles are not labeled in the figure legend and the identity of the sample remains unclear to the reader. The authors should include the appropriate labelling.

The missing label has been included in the respective figure legend.

2. Fig. 2C and D: The authors could add a scale to the Y axis.

The scale was added to indicate that the starting point for all traces is 1, otherwise these are arbitrary units.

3. In POST-1 the tRNA states are denoted as P/P and E/E. However, with the head of the SSU swiveled one would expect rather chimeric ap/P and pe/E states. If this is the case, the authors should correct their nomenclature.

Indeed, the head is swiveled and the tRNAs are in fact in ap/P and pe/E chimeric states. We have now updated Figure 6A and the text on pages 15 (results) and 18 (discussion) to reflect this.

4. Please clarify which CC is given in Table S1: CC_{mask}?, CC_{box}? Or other?

We now provide CC_{Volume}, CC_{Mask} and CC_{Box} in the updated Table S1.

Suggestions:

1. The biochemical data show convincing evidence for a role of eEF3 in controlling E site tRNA ejection. Interestingly, the authors discover a contact between the L1 stalk and eEF3 that appears to be based on charge complementarity. It is not so clear how this attracting interaction is used by eEF3 to facilitate opening of the L1 stalk and tRNA release as the authors propose. Perhaps eEF3 rather fosters an inward-facing, closed conformation of the stalk.

We recognize that we have neglected to emphasize that the L1 in position observed in the presence of eEF3 is not the same as the L1 in position observed in a POST-1 state that lacks eEF3. In the presence of eEF3, the L1 stalk moves slightly outwards, therefore, we have decided to rename the position as “int” for intermediate, this is now mentioned in the text on page 14. We have also included additional panels in Figure EV5A-D to highlight the difference between the L1-“in” and “int” conformations observed in POST-1 and POST-2, respectively.

The mechanistic model may be improved if the authors could include a speculation about the mechanism of stalk opening. Does ATP hydrolysis play a role, or ribosomal rotation?

A comparison of POST-2 with eEF3 and E-tRNA and POST-3 with eEF3 and no E-tRNA, suggests that rotation does not play a role since both are non-rotated. In our structures, the resolution is not sufficient to unambiguously determine if the individual states have ATP or ADP, a mixture of ATP and ADP, or possibly even ADP+Pi, therefore, we have refrained from making statements about the role of ATP in the function of eEF3. The published biochemical experiments on the eEF3 ATPase address multiple turnover ATPase activity uncoupled to translocation and thus the coupling between translocation, eEF3 cycle and ATP hydrolysis remains unclear (and is well beyond the scope of this manuscript). Also the effect of mutations in the ATPase domain do not provide detailed information on the coupling between the ATPase and translocation functions, except for showing that in the absence of ATP hydrolysis, eEF3 is blocked on the ribosome. Given the lack of compelling biochemical evidence, we would prefer not to discuss the role of ATP binding/hydrolysis in eEF3 action, but have added this as a remaining question to be addressed in the future (last sentence of Discussion).

2. The following sentence is confusing: "Analogous to PRE-1 and PRE-2, both PRE-3 and PRE-4 states also had density for the nascent polypeptide chain, but extending from the A site (rather than the P-site tRNA) into the ribosomal exit tunnel." It may be misinterpreted as the PRE-1 and PRE-2 states contain the nascent chain being bound to the P site tRNA, which is not the case. The authors should clarify this sentence.

The sentence has been corrected to clarify that all PRE states had nascent chains attached to the A-site tRNA and now reads "Analogous to PRE-1 and PRE-2, both PRE-3 and PRE-4 states also had density for the nascent polypeptide chain extending from the A-site tRNA into the ribosomal exit tunnel"

Referee #2:

Previous studies have suggested that the fungal-specific elongation factor EF-3 acts to stimulate coupled release of deacylated tRNA from the E site and binding of ternary complex (TC) to the A site; and structural analysis of an EF-3/80S complex revealed the EF-3 interaction surfaces on the back of the 40S and 60S subunits. This paper presents a combination of biochemical analysis of elongation in a purified system, ribosome profiling in EF-3 depleted cells and cryoEM analysis of various native EF-3 bound 80S complexes in an effort to better establish the mechanism of EF-3 function in the elongation cycle.

By monitoring rates of di- or tri-peptide synthesis in a fully reconstituted system, the results in Fig. 1C-D show that EF-3 has no effect on the rate of dipeptide synthesis, but it greatly accelerates tripeptide synthesis. The results in Fig. 1D suggest that EF-3 does not simulate the rate of TC binding to the A site following EF-2-catalyzed translocation of the dipeptidyl-tRNA to the P site. By following the rate of puromycin incorporation, the results in Fig. 2A suggest that EF-3 has relatively little effect on the rate of peptide bond formation. By directly monitoring the rate of dissociation of labeled E-site tRNA, the findings in Fig. 2C indicate that EF-3 promotes dissociation of E-site tRNA following EF-2 catalyzed translocation of dipeptidyl-tRNA, even when binding of TC to the A-site is not occurring (purple curve in Fig. 2C). The experiments in Fig. 2D indicate that

binding of TC alone following EF-2 catalyzed translocation is insufficient to evict the E site tRNA, but which occurs efficiently when EF-3 is also present. The authors conclude that EF-3 promotes release of E-site tRNA but has no role in stimulating TC binding to the A site, and that these two events are not coupled in the manner concluded from previously published experiments. Overall, these findings suggest that EF-3 stimulates the rate of elongation by 10-fold or more (depending on the codons involved) by promoting dissociation of E-site tRNA in a late stage of translocation. Ribosome profiling results in cells depleted of EF-3 revealed a depletion of elongating ribosomes with an empty A site (with 21nt RPFs) at the expense of those with occupied A sites (with 28nt RPFs) and the use of different antibiotics in the extracts led to the conclusion that the pre-translocation complexes following peptide bond formation dominate the 28 nt RPFs, consistent with a rate-limiting defect in translocation-which is in agreement with the biochemical data. Examining codon pause scores indicated a loss of pauses at non-optimal codons in the A site in the 21 nt RPF data; and a loss of pausing at Pro, Gly, and Asp codons in the P site in the 28 nt RPF data, upon depletion of EF-3. These results were attributed to translocation becoming rate limiting rather than A site decoding for poor codons, and presumably rather than a different step in translocation that is normally slow at P,G and D codons, upon EF-3 depletion. The third part of the paper describes cryoEM analysis of 80S ribosomes affinity purified from cells using TAP-tagged EF-3. A variety of different states were observed, some containing well-resolved EF-3 and others not. Analyzing the former shows EF-3 binding to the 40S and 60S subunits in a location and manner expected from the previous structural analysis of EF-3/80S complexes by Anderson et al. Among these, two represent post-translocation states based on the presence of peptidyl tRNA in the P site, but only one of the two contains an E site tRNA. Comparing these last two suggests that dissociation of E site tRNA is accompanied by movement of 60S protein L1 to an "out" configuration in which it is no longer engaged with the E tRNA. Two other complexes with well-resolved EF-3 contain A and P site tRNAs with the nascent peptide on the A site tRNA, signifying pre-translocation complexes, suggesting the EF-3 binds throughout the elongation cycle, even though it seems to affect only the late stage of translocation by helping to release the E site tRNA. Three other complexes containing poorly resolved EF-3 or no EF-3 density at all were also analyzed, two in pre- and one in post-translocation configurations based on the tRNAs present in the decoding sites. Interestingly, all of the latter feature intersubunit rotation, which is thought to occur during translocation; whereas the four complexes containing well-resolved EF-3 all represent non-rotated states. These findings suggest that stable EF-3 binding requires the non-rotated state of the ribosome, which is consistent with loss of EF-3 contacts with the 40S head predicted for the rotated 80S ribosome. Because two of the non-rotated, EF-3-bound complexes appear to be post-translocation complexes based on containing peptidyl-tRNA in the P site, the authors propose that EF-3 preferentially binds to, or stabilizes 80S ribosomes that have switched from rotated back to non-rotated subunit interfaces following translocation, and that EF-3 then catalyzes dissociation of E site tRNA from this non-rotated state by displacing L1 to its "out" conformation. This model envisions that EF-3 remains tightly bound to the non-rotated ribosomes through TC binding to the A site and peptide bond formation, binds weakly as intersubunit rotation proceeds and during EF-2 catalyzed translocation, and then resumes stable binding when the ribosome switches back to the unrotated state (or promotes this switch), where it catalyzes release of the E-site tRNA. The biochemical data, and ribosome-profiling data imply that EF-3 only significantly stimulates E-site tRNA release and does not accelerate TC recruitment to the A site or peptide bond formation even though it appears to be stably bound to the ribosome at all of these steps.

General Critique: There is a wealth of valuable data in this report obtained by a very interdisciplinary approach that draws on different strengths of the three labs involved, and which appear to significantly advance our understanding of EF-3 function. The data and final model depart substantially from the previous conclusion that EF-3 catalyzes a coupled release of E site tRNA and A site binding of TC with evidence that these events do not exhibit obligatory coupling and that EF-3 can stimulate tRNA dissociation from the E site in the absence of TC, and no apparent acceleration of TC binding to the A site given by EF-3. The ribosome profiling data support a defect in translocation that would be expected from defective E-site tRNA dissociation, and the cryoEM structures provide evidence consistent with EF-3 catalyzing dissociation of the E-site tRNA without the presence of TC in the A site.

The manuscript is difficult to read however and the presentation could be considerably improved to explain things better, improve the figures, and eliminate incorrect or confusing statements in all three sections of the paper.

We thank all the reviewers for the detailed comments and criticisms, which we have carefully addressed, and believe that this has considerably improved the manuscript.

In addition, while the profiling data provide evidence that EF-3 depletion alters the rate-limiting steps in elongation, evidence is lacking that EF-3 depletion actually reduces the rate of translation at the elongation stage in vivo. For some of the analyses of ribosome profiling data, it is unclear whether biological replicates provide consistent results and support the derived conclusions. And there are deficiencies in the analysis of bulk polysomes and polysome association of EF-1A and EF-2 upon EF-3 depletion from cells.

We have worked hard to address the reviewer's concerns. First, we have indeed performed replicates of the ribosome profiling experiments (we apologize for not being clearer in the original manuscript) and these results are internally consistent (see new panels 3B and 3D in in Figure 3). In addition, as the reviewer suggests, while the profiling data suggest that there are new rate limiting steps in elongation, a clear way to identify elongation effects is with a run-off experiment. We have now performed polysome run-off experiments, including two biological replicates for each condition to support our conclusions. Finally, we have repeated the polysome profiles with equal amount of total RNA input and internal control (ribosomal protein) to show that the association of elongation factors (i.e. eEF1A and eEF2) with polysomes are unaffected by changes in eEF3 concentrations in the cell.

There are also questions about whether appending the tag to EF-3 alters its function in vivo, and whether the complexes analyzed by cryoEM were all truly enriched for EF-3, as an untagged control strain was apparently not employed in the purification experiments.

We think it is unlikely that the C-terminal TAP tag has any adverse effect on EF-3 function since when eEF3 is bound to the ribosome, the C-terminus is located far from the ribosome (new Appendix Figure S2A-B) and we observe no density for it suggesting no defined interactions. Moreover, if the TAP-tag inactivated eEF3, the cell would not be viable since the eEF3 functionality is essential for survival. Nevertheless, we have also performed growth assays comparing the growth of the wildtype and TAP-tagged eEF3 strain at different temperatures and observed no influence on the TAP-tag on fitness (new Appendix Figure S2C-E). To address the second point as to whether the complexes were truly enriched for

eEF3, we have now performed the requested control experiment, where we repeated the Tap-purification but using an untagged control strain in parallel with the tagged strain. As can be seen in the new Appendix Figure S2F, no ribosomal complexes are purified from the untagged strain, unlike in the TAP-tagged strain. These additional experiments are now mentioned briefly on page 11.

Specific comments:

-The description of the kinetic data in Figs. 1-2 is rather dense, and would benefit from a more expansive treatment indicating more clearly the specific question being addressed by each experiment regarding whether a particular step in elongation is stimulated by EF-3, and the features of the assay that allow that step to be studied in isolation of the others, followed by a succinct conclusion reached from the findings before moving on to the next partial reaction under consideration. The next five comments are all related to this one.

We thank the reviewer for the detailed suggestions, which we implemented in the text on p. 5-8 as detailed below.

-Fig. 2A could be improved and a detailed explanatory legend provided that better matches the narrative in the text. It does not do a good job of depicting the different stages of translocation as described in the text on the top of p. 7. CCW and CW are not defined in the legend, nor are PRE and POST; and the hybrid and "chimeric" states are not indicated in the figure. The three steps described in the text are not mentioned explicitly in the figure. Whether the tRNAs are in the P/P or P/E states could also be shown.

Figure 2A has now been amended to better match with the narrative in the text.

-It would be useful to add a sentence towards the end of p. 7 giving the conclusion about the data in Fig. 2B regarding the importance of EF3 for the rate of translocation, ie. that EF3 is dispensable for efficient translocation but enhances the rate of the reaction.

We note that eEF2 promotes only partial translocation, as deacylated tRNA is not released from the E site unless eEF3 is added. This is now clarified along the lines suggested by the referee on p. 8.

-It might be helpful to add to Fig. 2A a schematic of a reaction depicting E site clearance in the absence of TC-Val, which EF-2 and EF-3 can catalyze based on the purple curve in Fig. 2C.

Figure 2A has now been amended to depict E-site clearance in the absence of TC-Val.

-p. 9: the sentence "We further tested whether the tRNA dissociation from the E site and TC binding to the A site are directly coupled." might be improved by writing "We further tested whether the tRNA dissociation from the E site and TC binding to the A site are directly coupled in the absence of EF-3". The sentence "These experiments clearly establish that binding of the cognate TC into the A site is not coupled to the dissociation of the E-site tRNA" might be improved by writing "These experiments clearly establish that binding of the cognate TC into the A site alone following EF-2 catalyzed translocation, is not sufficient to promote dissociation of the E-site tRNA"

We changed the first sentence mentioned by the referee to better describe a broader question asked in these experiments (this is the introductory sentence for the data presented in Fig. 2D, E). We followed the suggestion of the referee for the second sentence.

-The authors conclude that EF-3-catalyzed dissociation of E-site tRNA is not coupled to TC binding to the A site; however, is it possible that the more rapid and complete dissociation of E-site tRNA observed in the presence versus absence of TC shown in Fig. 2C (green vs purple curves) is an indication that coupling of the two reactions increases the efficiency of EF-3 function?

We have no indication for such effects, as the simplest explanation for the difference in the end level for tRNA dissociation time courses with $eEF2+eEF3 \pm TC$ (Fig 2C) is the second-round translocation after Val-tRNA binds and the tripeptide Met-Phe-Val is formed. As expected, the tRNA dissociation rate observed in the presence of TC-Val, eEF2 and eEF3 is not monophasic and the reaction is overall somewhat slower than in the absence of TC, which reflects additional time required for TC-Val binding, tripeptide formation and the second translocation. As there is no significant acceleration of eEF3 by TC-Val, we thought that the explanation of the end level differences by the second translocation round is the most plausible one.

-p. 9: indicate that the inhibitors used in ribosome profiling are added to the cell extracts to prevent interconversion between the states shown in Fig. 3A following cell lysis and thereby help to preserve the distribution among the states present in vivo.

This is an important point. We have clarified in the text at the bottom of page 8 that the elongation inhibitors were added to the cellular lysates to prevent movements between ribosome states post cellular lysis.

- p. 10: For the sentence "By contrast, in eEF3d cells, we observed a drastic reduction in 21 nt RPFs (and a concomitant increase in 28 nt RPFs) when compared to WT cells (Figures 3B and Appendix Figure S3C), indicating the accumulation of PreTrans state ribosomes." Isn't it more correct to say that based on these data there is an accumulation of ribosomes in either the PrePT or PreTrans states? The next experiments with anisomycin are used to distinguish between these two states by blocking conversion of PrePT to PreTrans states in the extract, which allows accurate quantification of the PrePT state in vivo. If my interpretation of this is correct, it would be helpful to introduce the anisomycin experiments in Figs. 3C-D in that way.

The reviewer is correct on this point and we have revised the text to increase clarity.

-p.10: the sentence "We further found that the footprints from eEF3d cells are almost devoid of 21 nt RPFs when compared with the WT samples prepared with CHX+ANS (Figure 3D)." should be followed by an interpretative comment to the effect that because 28nt RPFs originate solely from PreTrans 80S complexes owing to dissociation of PrePT 80S ribosomes when ANS is present, these results indicate an overabundance of PreTrans complexes on EF-3 depletion from cells, consistent with impaired translocation rather than a defect in peptide bond formation in vivo.

Again the reviewer is correct and we have included an additional sentence to clarify this point.

-Fig. 3A-D and associated text: It's unclear whether the results shown here were obtained reproducibly in biological replicates. Ideally, the mean proportions of RPFs of different fragment lengths should be presented with errors and statistical analysis from replicate experiments. At least, curves like those in panels B and D should be shown for each of the replicates.

We apologize for not adequately documenting the nature of our data. We have (and had already) generated biological replicates for all the ribosome profiling experiments (wild type and eEF3-depleted cells prepared with CHX+TIG or CHX+ANS). In particular, in the replicate experiments we observed a reproducible and significant reduction of 21 nt RPFs in eEF3-depleted cells. We have now included two biological replicates in Figure 3B (for CHX+TIG) and 3D (for CHX+ANS) in the updated manuscript.

-Fig. S3D: It should be stipulated whether CHX was added to cells to freeze elongation in these experiments. (In fact, there are no Methods given for this experiment or the others in Fig. S3.) Also, the Western signals for EF-1A and EF2 on polysomes are lower in the mutant vs. WT. To conclude, as they do, that association of EF-1A and EF2 with polysomes is normal on EF-3 depletion, the western signals would have to be normalized to the A260 tracings, or to Western signals for a ribosomal protein, and results presented from replicate experiments. It would also be reassuring to see that the polysomal signals for these factors is eliminated on RNase treatment to collapse polysomes to monosomes to rule out non-polysomal aggregates contributing to the polysome signals.

The reviewer is correct in pointing out that unequal amounts of input material for the polysome profiling made interpretation of this experiment confusing. We have now performed polysome profiling experiments (with CHX added during cell lysis to stop translation) using equal amount of total RNA (determined by A₂₆₀) and have included ribosomal protein RPL4 as an internal control for polysomes. A detailed description of these experiments has been added to the Methods section. In this set of experiments, we observed that the association of eEF1A and eEF2 with polysomes remains unaffected upon eEF3 depletion (see new Figure EV2D).

-Assuming CHX was added to cells for the experiment in Fig. S3D, it is surprising that the polysomes are depleted in the mutant rather than accumulating and showing a shift to larger polysomes in the manner expected for reduced rates of elongation. Rather, polysomes relative to monosomes are depleted, indicating reduced initiation, or wholesale mRNA degradation, which could occur as a secondary response to 8 h of EF-3 depletion. It seems important to compare polysome profiles in the presence and absence of CHX, as a reduced rate of elongation in the mutant should diminish polysome run-off in the absence of CHX compared to what occurs in WT cells, which would provide direct evidence for reduced elongation rates in vivo on depletion of EF-3.

As suggested by the reviewer, we have performed polysome profiling experiments with CHX (see the point above) and without CHX (polysome run-off). In the run-off experiments, we observed reproducible increased polysome-to-monomosome ratio in eEF3 depleted cells (see new Figure EV2C). These additional experiments (polysome profiling with and without CHX) provide further support for the cryo-EM data (Figure 7) and the ribosome profiling data (Figures 3B and 3D).

-p. 11 bottom and Figs. 3G-H: It's difficult to comprehend the loss of P site pausing at P, G, and D codons as determined by analysis of 28nt RPFs in the experiments of Fig. 3G and H. Since CHX + ANS enriches for PreTrans 80Ss, the implication appears to be that translocation, rather than peptide bond formation, is slow with these P site codons in WT cells. But if EF-3 is slowing down translocation, one might have expected exacerbation of the slow translocation rates at these codons. Do the authors have to postulate that a different step in translocation is impaired which becomes more rate-limiting on EF-3 depletion distinct from the aspect of translation that is slow in WT cells at these codons? A more illuminating interpretation of these results is needed.

This is absolutely how we interpret this experiment and have tried to clarify this in the text.

-the legend to Fig. S3E doesn't stipulate whether A or P site pauses are being represented, and other details about how these motifs were determined are lacking; and if they were observed in both replicates.

We have performed the requested analysis for two biological replicates (see new Figure EV2E) and both replicates show similar motif signatures of ribosome pausing at P, D, and G codons. In addition, we have made clarifications in the legend to indicate which sites are being characterized. A description of how the logos were generated has been added to the Methods section.

-Fig. 4A doesn't stipulate what antibody is used in the Western blot. One would like to see that the supernatant lane would be essentially blank from a parallel pull-down of the WT untagged strain to be sure that the recovery of this band and associated RPs is dependent on the EF-3 pull-down. This seems important since the presence of EF-3 is either questionable or lacking in the complexes shown in Fig. 6A, C, D but it is assumed that they contained EF-3 during the pull-down.

As mentioned above, we have now performed the requested control experiment, where we repeated the Tap-purification but using an untagged control strain in parallel with the tagged strain. As can be seen in the new Appendix Figure S2F, no ribosomal complexes are purified from the untagged strain, unlike in the TAP-tagged strain. This is mentioned on page 11.

-Figs. S7E-G don't do a good job of showing the predicted clash of the EF-3 domain with the ribosome for a predicted open conformation. A blow-up of the predicted clashing elements seems necessary. Also the predicted dissociation seems to be restricted to the cyan HEAT domain, the only component of the ABC1-HEAT-4HB domain that contacts the 40S, which would be helpful to point out more explicitly.

As requested, we have now included new enlargements panels so that the reader can better see the predicted clashes between domains of eEF3 and the ribosome when adopting an open conformation. This is now in Figure EV3E-J and cited on page 12.

-p.11: "...the N-terminal HEAT and 4HB interact with the 40S subunit, whereas the C-terminal CD contacts exclusively the 60S subunit." This sentence is misleading in implying 4HB interactions with the 40S.

This sentence has been corrected and reads now "the N-terminal HEAT interacts with the 40S subunit, whereas the C-terminal CD contacts exclusively the 60S subunit"

-p13-14: Indicate whether the specific contacts of EF3 HEAT, CD, and ABC2 domains with the ribosome mentioned here were observed previously in Anderson et al, or instead newly observed here.

The previous reconstruction from Anderson et al was at 9.9A and allowed only a rigid body fit of the X-ray structure of EF3 into the map and at the time there was not even a complete model for the yeast 80S ribosome, therefore, only a very global model of the interactions was provided. All the details described here are therefore new. We have now added the sentence to make this clear..." With the improved resolution compared to the previous eEF3-80S structure (Andersen et al., 2006), details of the interactions of the eEF3 with the ribosomal components can be more accurately described."

-Analysis needs to be provided to show whether appending the TAP tag to EF-3 affects cell growth or polysome profiles.

As mentioned above, we have monitored cell growth at three different temperatures for the wildtype and eEF3-TAP tagged strain and observe no differences (new Appendix Figure S2C-E). This is mentioned on page 11.

-p. 18: I take issue with the sentence "Similar to previous studies where eEF3 was depleted in the cell (Kasari et al., 2019a), we observe general defects in translation elongation." as the polysome profiling shown in Fig. S3 reveal an initiation defect and they have not provided evidence that the rate of elongation is reduced in vivo on EF-3 depletion. The ribosome profiling data indicate that EF-3 depletion appears to be altering the rate-limiting steps in the elongation cycle, but they don't indicate directly that the rate of translocation is reduced from its WT rate and that this reduces the overall elongation rate. As such, using the word "trapped" in the following sentence is not justified.

As suggested by the reviewer, we have presented more concrete data from polysome run-off experiments to show that depletion of eEF3 results in defective translation elongation (Figure EV2C). Together with our in vitro biochemistry and ribosome profiling results, we believe that our data support the conclusion that depletion of eEF3 traps ribosomes in a pre-translocation state.

-p. 18, regarding the sentence: "A strong prediction of the biochemical and ribosome profiling results is that the ribosomes that accumulate in cells on eEF3 depletion would contain three separate tRNAs." shouldn't Fig. 7H be cited here showing this aberrant species? Also, shouldn't this aberrant state be generated from the pre-translocation intermediate in panel D vs. the post- intermediate in panel E with EF-2 in the A site? Later on p. 19, the sentence " we envisage that ribosomes are trapped in the POST-2 state with deacylated tRNAs in the E- and P-sites and a peptidyl-tRNA in the A site." seems to indicate that the state in Panel H would originate from that in Panel F. Either the thinking or the writing on this point seems to be unclear.

The reviewer is correct that the arrow directly from the panel E to H is somewhat confusing. The arrow should really come from the state AFTER EF2 has left but before eEF3 binds (because it cannot since there is no eEF3). Therefore, we have re-arranged the scheme so that we have now two arrows after POST-1, one for the situation where eEF3 is present and one where it is depleted. Note that the product of EF2 acting on POST-1 is POST-2,

regardless of whether eEF3 is present or not, which we mention in the legend. We have also labelled panel H as POST-2 + A-tRNA, which we hope also makes things clearer.

-Fig. 7 is confusing in that two arrows are not shown between panels G and A representing TC binding to A site (by yellow tRNA shown) and peptide bond formation (arrow missing) that switches the peptidyl-tRNA from the P site (red) to A site (yellow). Also wording at the end of the legend is awkward and confusing.

As requested, we have now included two arrows between G and A to represent ternary complex binding and peptide bond formation, and modified the legend text accordingly.

-p. 18-19: in the phrase "whereas dissociation of the factor and back swivel occur in a later step (Figure 7A-B)." shouldn't Fig. 7F-G be cited instead of A-B?

The back swivel occurs upon eEF2 dissociation, i.e. between Figure 7E and 7F and not 7A-B as was incorrectly stated. This has now been corrected in the text.

Referee #3:

Ranjan et al. address the yet vaguely defined function of the translation elongation factor eEF3, which is essential in fungi but absent in most other eukaryotes. Using a fully reconstituted translation elongation system from the bona fide model organism *Saccharomyces cerevisiae*, Ranjan et al. show that eEF3 is responsible for E-site tRNA ejection independent of other translation elongation factors. Ribosome profiling in eEF3-depleted bakers' yeast suggests a general role for eEF3 in translation elongation. Ex vivo cryo-EM structures at 3-4 Å resolution partially complement the known translation elongation cycle structurally and allow insights into the molecular mechanism of eEF3. It is important to note that the X-ray crystal structure of yeast eEF3 (Andersen et al. 2006) has been used to fit the cryo-EM map, which complements a study on New1p (Kasari et al. 2019 NAR). Unfortunately, the authors could only visualize eEF3 at 80S in a single conformation after glutaraldehyde crosslinking and trapping by non-hydrolysable AMPPNP, so both the chemomechanical coupling and the function of the two nucleotide-binding sites of eEF3 remain unknown.

The manuscript by Ranjan et al. is generally well written and very interesting for the scientific community. The data are experimentally diverse, supplement each other and are presented in a comprehensible way. However, some concerns need to be addressed to support the overall conclusions of the authors.

Major issues:

1) Biochemical in vitro studies: The authors do not state that (or how much) ATP was used in the in vitro experiments. This point is critical because the ATP-occluded state has been described as ribosome-bound in this and previous studies. In fact, if ATP was not present, eEF3 might have used GTP from the TC preparation. In this case, the authors should show in additional experiments (e.g. like Fig. 1C, D and 2C) that ATP does not alter the outcome. In addition, they could provide ATPase and GTPase assays (colorimetric or radiographic) to show that eEF3 can utilize GTP in a similar way as ATP, which might well be true for an ABC-type system. Since eEF3 has been used in excess, even a low, unspecific: GTPase activity could suffice to accelerate the reaction, but the kinetic values would not reflect the physiological situation, which should be

clearly stated. If control experiments with ATP have to be repeated, I suggest that the authors additionally use ADP and AMPPNP to correlate the function of eEF3 in the translation cycle with its ATPase activity, similar to what has already been done for Rli1/ABCE1, for example.

All experiments where eEF3 is present were carried out in the presence of ATP; we apologize for the unintended omission. This information has been now included in Materials and Methods, p. 39. We did not intend to study the role of ATP hydrolysis in this paper and it would be difficult to do so, because eEF3 can bind and hydrolyze not only ATP, but also GTP which is presented in every experiment at a rather high concentration.

2) Ribosome profiling studies: The authors should exclude that the levels of eIF5A have dropped significantly upon eEF3 depletion. Given a half-life of 9.1 h (source: <https://yeastmine.yeastgenome.org>) and impaired translation, the effects seen in the ribosome profiling experiments after 8 h may reflect partial loss of eIF5A. Especially, since the codon-resolved analysis showed an overlap with eIF5A function. The authors could easily verify this by using the samples from Fig. S3A for an immunoblot against eIF5A.

We appreciate the reviewer's concern. If the level of eIF5A had dropped upon eEF3 depletion, we would have expected to see increased ribosome pausing at eIF5A-sensitive motifs (i.e. P, G, and D codons). Given that we observed reduced pausing at eIF5A-sensitive motifs, we think that a reduction in eIF5A protein levels is unlikely.

3) Structural data - Page 17-18 and Fig. 7:

„This is somewhat surprising since formation of hybrid states in higher eukaryotes is generally assumed to be concomitant with intersubunit rotation (Behrmann et al., 2015; Budkevich et al., 2011; Svidritskiy et al., 2014), however, we note that subunit rotation and tRNA movement is only loosely coupled on bacterial ribosomes (Fischer et al., 2010). Since rotation of PRE- 2 would result in a state similar to PRE-3, we suggest that subunit rotation is what triggers the low affinity form of eEF3, effectively leading to unstable ribosome binding.”

„The ability to capture four different PRE and three different POST states of the ribosome by co-immunoprecipitation of tagged-eEF3 suggests that eEF3 is omnipresent during translation elongation in yeast...”

„(C) Rotation of the ribosomal subunits leads to unstable binding of eEF3 to a ribosome with A/A- and P/E-tRNA (PRE-3) as well as to (D) a fully rotated ribosomal species bearing hybrid A/P- and P/E-tRNAs (PRE-4)”

The authors should discuss the point that usage of AMP-PNP and a chemical crosslinker might stabilize non-physiological states of the 80S-eEF3 complex or even complexes that are never formed in vivo. This is reminiscent of the situation with ABCE1 in translation initiation complexes, which in structural studies is always bound to 40S subunits in the presence of AMP-PNP (see e.g. Simonetti et al., 2020 or Heuer et al., 2017), but dissociates upon ATP addition (see e.g. Simonetti et al., 2020). A recent BioRxiv preprint (Kratz et al., 2020) now demonstrates ABCE1 in initiation complexes in a new state, possibly manifesting its role in translation initiation, but certainly showing that AMP-PNP-containing samples cannot be used to extend structural data to functional knowledge without validation by biochemical experiments. If this cannot be ruled out by convincing arguments or references, the authors should relativize the respective statements. Alternatively, cryo-EM experiments (low resolution should be sufficient)

could be performed using ex-vivo samples without AMP-PNP. The authors could also consider biochemical (e.g. co-immunoprecipitation) or biophysical (e.g. fluorescence-based) eEF3-binding studies of defined 80S complexes (in the presence of antibiotics) to support their statements.

There appears to be a misunderstanding as to how the complexes were prepared. The cells were not lysed in the presence of AMP-PNP, nor were they applied to the beads or washed in the presence of AMP-PNP ...only the elution (by protease) of the complexes was performed in a buffer containing AMP-PNP (as stated in the methods on page 40). Since the lysing and application procedure takes over 3 hours, we think that it is unlikely that complexes will be present that were never formed in vivo – more likely that complexes will have disassembled during this time that we don't observe (perhaps explaining why we do not observe for example the ternary complex delivery state; see the penultimate comment of reviewer #2 above). Moreover, we have also performed the elution without the presence of AMP-PNP and observed similar results (now included as new Appendix Figure S2F), which suggests that the purification of ribosomal complexes does not require AMP-PNP. Collectively, we suspect that the molecule we see in the active site is ATP and not AMP-PNP however, we do not have the resolution to distinguish between these two molecules and are thus careful not to favor one over the other in the text, for example, as on page 12 "We note that electron density consistent with ATP (or ADPNP) was observed in the active sites of the ABC1 and ABC2 nucleotide".

Minor points

1) Fig. S1A (left panel): The authors should state what "nat" means.

We have included the meaning of "nat" (native proteins) in the respective figure legend.

2) Fig. S2A: Please indicate how much 80S IC was applied to directly correlate the 2 μ M and 4 μ M eEF3 concentration.

We have included the concentration of 80S IC and ternary complexes used in Figure EV1A in the respective figure legend.

3) Fig. 1D: What are the brown triangles? Puromycin test as in Fig.2?

The description is now added in the respective figure legend. Brown triangles show the lack of reaction in the absence of eEF2 and eEF3.

4) Page 6 and Fig. 1D „(ii) eEF2 alone can complete the translocation process given enough time" : The Met-Phe-Val formation is not complete with eEF2 alone, which the authors also state on page 9: „eEF2 alone promotes partial tRNA translocation" : In Fig. 1D some data points are missing at the end of the red trace. The authors should comment on why they were removed or not analyzed in this experiment. The authors should clarify their message and state that translocation by eEF2 alone is either slow but complete and the difference is within the range of error or it is slow and incomplete.

The time range has been now adjusted to be identical in all experiments shown of Figure 1D. We initially made a longer time course for the +eEF2+eEF3 experiment, but then realized that the long time points are not informative, as saturation is reached at 100 s with both eEF2 alone and eEF2+eEF3.

5) Page 7 and Fig. 2B: „We note that in the presence of eEF3 and eEF2, the reaction with Pmn was faster than with eEF2 alone, suggesting that binding of eEF3 to the ribosome stabilizes a ribosome conformation that is somewhat more active in peptidyl transfer reaction with Pmn, possibly by stabilizing peptidyl-tRNA in the exit tunnel": However, the reaction is incomplete (app. 75%) in the presence of both factors eEF2 and eEF3 (green) compared to eEF2 alone (red). Thus, eEF3 seems to have an inhibitory role on a portion of ribosomes during the Pmn reaction, which should be mentioned here. Interestingly, the end point defect is comparable to the one in Figure 1D, which may indicate heterogeneity in the 80S 2C population.

Pmn reaction is only a diagnostic test for the movement of the peptidyl-tRNA to the P site. We do not know how the factors affect Pmn binding (as opposed to the chemistry step). A detailed study of why there are moderate differences in the Pmn reaction with different factors is beyond the scope of the paper, as the important information is that there is Pmn reaction when the factors are present, as opposed to no reaction in the absence of the factors. An additional argument that the A-site peptidyl-tRNA has moved to the P site is provided by the experiment showing the tripeptide formation upon addition of eEF2 and TC-Val.

6) Fig. S2G: The labeling is a bit irritating. It looks like only eEF3 was added because in all previous figures, all the components were listed. I suggest removing eEF3 here, leaving just the tRNA labels or adding TC and eEF2 to be consistent with the other figures.

We have re-labeled the Figure as suggested by the reviewer.

7) Page 8 and Fig. 2C: „When a similar experiment is carried out in the absence of TC-Val, only the first translocation can occur; we observe that in this case, the fluorescence of tRNA^{Met} decreases considerably, but not to the same extent as when TC-Val is included (Figure 2C, purple trace)": The authors should consider 80S 2C heterogeneity as a reason for the difference in E-site tRNA clearance in the absence and presence of the TC. If ATP was not present in the reaction mixture, the absence of GTP could be the reason for the incomplete reaction of the purple trace.

We are really sorry for forgetting to mention the presence of ATP in our experiments, which is now indicated in Materials and Methods. We have no reason to assume a heterogeneity of 80S 2C, because the end level of tripeptide formation is the same after incubation of eEF2 or eEF2+eEF3. For the deacylated tRNA that moved to the E site in the presence of eEF2 and eEF3 (in the absence of TC-Val) there are three possible scenarios: (1) tRNA^{Met} moved uniformly on all ribosomes from the P to the E site, but did not dissociate; the end level of the fluorescence time course in Fig. 2C reflects the characteristic fluorescence of the tRNA bound to the E site; (2) tRNA^{Met} moved uniformly on all ribosomes, but the affinity of tRNA^{Met} to the E site is low, so that a fraction of it dissociates from the ribosome, whereas part remains bound, as defined by the K_d value; the resulting fluorescence is a mixture of tRNA bound to the E site and tRNA free in solution; (3) tRNA^{Met} moved to the E site on only part of the ribosomes, so that the end level of the time course in Fig. 2C is a combination of the tRNA in the P site and the tRNA dissociated from the ribosome. We can exclude point (3), because if tRNA^{Met} were not removed from the P site, the A site would remain occupied as well and thus TC-Val were not able to bind. This is, however, not the case, as shown in Fig 1D. The new experiment shown in Fig 2D seems to favor scenario 2, because there is physically less tRNA^{Met} coeluting with ribosomes, but this may be due to a significant dilution upon gel filtration and it is thus unclear whether this is also the case for the reaction conditions of Fig

2C. Thus, both scenarios 1 and 2 are possible and we added a few comments explaining this on p. 7-8.

8) Page 9: The study by Velechano & Alepuz (NAR, 2017) should be mentioned in the context of the global role of eIF5A in translation.

The study has been cited.

9) Page 13 "archaeal 40S-ABCE1 complex (Heuer et al., 2017)":

The archaeal complex with the highest resolution (2.8 Å) obtained for SSU-ABCE1 (which is a 30S ribosomal subunit in Archaea, not a 40S) was described in Nürenberg-Goloub et al., 2020, EMBO J. Heuer et al. 2017 NSMB describes the yeast 40S post-splitting complex. The author might consider to include an additional reference for the open conformation of ABCE1 (Barthelme et al. 2011 PNAS).

We have compared the archaeal 30S-ABCE1 from Heuer et al 2017 with the highest resolution yeast 40S-ABCE1 from Nürenberg-Goloub 2020 and they are very similar to each other. Nevertheless, we have updated the figure and citation to the more recent 30S-ABCE1 structure since as the reviewer states, it has higher resolution. With respect to the closed conformation, the figure is actually the open conformation from Barthelme et al 2011 as stated in the legend. The citation in the text has been corrected to reflect this. We have also compared the Karcher and Barthelme open conformations and they are also very similar.

10) Page 13: "...ribosome-stimulated hydrolysis of ATP to ADP would promote eEF3 dissociation": The authors should provide a reference, which describes the stimulation of eEF3 ATPase activity by the ribosome and, putatively, its dissociation from the ribosome upon ATP hydrolysis. Alternatively, they could provide examples from other ABC-proteins (ABCF? ABCE?) showing such behavior to support their statement.

We have included here the reference to the comprehensive review on eEF3 highlighting the studies that show ribosome-dependent stimulation of the ATPase of eEF3 as well as mutants that abrogate ATP hydrolysis and trap eEF3 on the ribosome.

11) Fig. S6D: The authors clearly present the presence of a nucleotide, which is most likely AMPPNP based on sample preparation and overall conformation of eEF3. However, ABC proteins essentially occupy Mg²⁺ ions with each nucleotide. The authors should show the density for the ions in this figure or comment on why the density is absent from their sample.

As discussed above, we suspect that it is probably ATP, rather than AMPPNP, but we do not have the resolution to distinguish between them. Nevertheless, in both cases we would expect a Mg²⁺ ion to be coordinated with the ATP/AMPPNP molecule, as pointed out by the reviewer. At this point, we need to re-emphasize that we do not have the resolution to see coordinated Mg ions, however, if we align ATP and ADP molecules with the coordinated Mg²⁺ ions from other structures, we see a very nice fit of the Mg²⁺ ion into our cryoEM map density. This procedure, also supports our interpretation that the density is consistent with ATP/ADPNP rather than ADP, consistent with the closed and not open conformation of the ABC domains. We have now included this analysis in Appendix Figure S5 panels F-K. However, we would like to again re-emphasize that the density we look at is a multibody map that combines density from all stable eEF3-containing states i.e. is a mixture of PRE-1, PRE-2, POST-2 and POST-3. While this suggests that all these states are also most likely ATP/ADPNP, consistent with their closed conformation, we have to acknowledge that the resolution of the individual maps does not directly allow such a conclusion. For this reason,

we are reluctant to make conclusions about the ATP/ADP state of each state and the role of ATP in the function of eEF3, especially with respect to the structural aspect of our study.

Thank you for submitting your revised manuscript, we have now received the reports from the three initial referees (see comments below). I am pleased to say that they overall find that their comments have been satisfactorily addressed and now support publication. Referee #3 still raises some issues regarding ATP/GTP concentrations and ADP-PNP binding to eEF3. Please carefully review these points and revise the manuscript as needed, as well as providing a brief point-by-point response to these two concerns when submitting the final revised version of the manuscript. In addition, I would like to ask you to also address a number of editorial issues that are listed in detail below. Please make any changes to the manuscript text in the attached document only using the "track changes" option. Once these remaining issues are resolved, we will be happy to formally accept the manuscript for publication.

REFEREE REPORTS

Referee #1:

The authors addressed all raised concerns and the revised manuscript is an excellent candidate for publication in EMBO Journal.

Referee #2:

I am fully satisfied with the additional data and revisions of text made to address my comments.

Referee #3:

eEF3 Revision

The authors have addressed all critical points except for two major aspects: 1) Biochemical and biophysical in vitro studies; 2) Structural data - page 17-18 and Fig. 7. Overall, the revised manuscript has been largely improved, and addressing these two pending issues would strengthen its conclusions.

Major concern 1 - biochemical and biophysical in vitro studies:

The authors do not state that (or how much) ATP was used in the in vitro experiments. I assume that they simply forgot to mention it. This point is critical, because the ATP-occluded has been described as ribosome-bound in this and previous studies.

The authors replied: All experiments where eEF3 is present were carried out in the presence of ATP; we apologize for the unintended omission. This information has been now included in Materials and Methods, p. 39.

Remaining concern: However, the ATP concentrations used in the kinetic experiments were not physiological and unexpectedly low, i.e. the authors used a 10-fold molar excess of ATP (20 or 40 μM) relative to eEF3 (2 or 4 μM). In the presence of ribosomes, eEF3 hydrolyses approximately 900-1000 ATP/min (kcat; Sarthy et al., 1997; Uritani and Miyazaki, 1988). If we now assume that only 0.35 μM (concentration of ribosomal complexes) of eEF3 exhibit the ribosome-activated ATPase activity, the total ATP would be hydrolyzed after 4 to 8 sec. This may not be critical for experiments with very fast kinetics (Fig 1B, C) but everything beyond this time span runs on GTP (Fig. 1D, 2C, E, especially 2E, EV1). Given the ATP vs. GTP concentrations in the cell, this is not physiological.

In their reply, the authors mention a 'rather high' GTP concentration, but how high is the concentration exactly? The authors should discuss these important experimental details in the main text. Mentioning the ATP concentration in Material & Methods only is, in my opinion, not sufficient. I encourage the authors to repeat the above-mentioned critical experiments in the presence of a higher ATP concentration. Alternatively, they should find convincing evidence to legitimate a 'GTP-fueled' kinetic analysis of a translational ATPase and mention it in the main text.

Major concern 2 - structural data, page 17-18 and Fig. 7:

The authors should discuss that the use of AMPPNP and a chemical crosslinker might stabilize non-physiological states of the 80S-eEF3 complex or even complexes that are never formed *in vivo*. This is reminiscent of the situation with ABCE1 in translation initiation complexes, which in structural studies is always bound to 40S subunits in the presence of AMP-PNP (see e.g. Simonetti et al., 2020 or Heuer et al., 2017), but dissociates upon ATP addition (see e.g. Simonetti et al., 2020). A new BioRxiv preprint (Kratz et al., 2020) now shows ABCE1 in initiation complexes in a new state, possibly manifesting its role in translation initiation, but certainly showing that AMP-PNP-containing samples cannot be used to extend structural data to functional knowledge without validation by biochemical experiments. If this cannot be ruled out by convincing arguments or references, the authors should relativize the respective statements. Alternatively, cryo-EM experiments (low resolution should be sufficient) could be performed using *ex-vivo* samples lacking AMP-PNP. To substantiate their statements, the authors could also consider biochemical (e.g. co-immunoprecipitation) or biophysical (e.g. fluorescence-based) eEF3-binding studies on defined 80S complexes (in the presence of antibiotics).

The authors replied: There appears to be a misunderstanding as to how the complexes were prepared. The cells were not lysed in the presence of AMP-PNP, nor were they applied to the beads or washed in the presence of AMP-PNP ...only the elution (by protease) of the complexes was performed in a buffer containing AMP-PNP (as stated in the methods on page 40). Since the lysing and application procedure takes over 3 hours, we think that it is unlikely that complexes will be present that were never formed *in vivo* - more likely that complexes will have disassembled during this time that we don't observe (perhaps explaining why we do not observe for example the ternary complex delivery state; see the penultimate comment of reviewer #2 above). Moreover, we have also performed the elution without the presence of AMP-PNP and observed similar results (now included as new Appendix Figure S2F), which suggests that the purification of ribosomal complexes does not require AMP-PNP. Collectively, we suspect that the molecule we see in the active site is ATP and not AMP-PNP however, we do not have the resolution to distinguish between these two molecules and are thus careful not to favor one over the other in the text, for example, as on page 12 "We note that electron density consistent with ATP (or ADPNP) was observed in the active sites of the ABC1 and ABC2 nucleotide".

Remaining concern: The authors included an SDS-PAGE/Coomassie showing that protease elution of eEF3 from the beads during immunoprecipitation in the absence and presence of ADPNP yields the same proteins. Indeed, this excludes the elution of proteins that were not bound to eEF3 in the absence of ADPNP. But why was ADPNP then added during elution? This is somewhat surprising as the authors could have performed the cryo-EM analysis on a native sample but decided to add an artificial nucleotide. Further, the addition of ADPNP during elution might have altered the conformation of eEF3 and its affinity for certain ribosomal states. For example, it may well be that eEF3 stably binds to the states PRE3 to POST1 (Figure 7) in a hybrid ATP/ADP state (similar to 80S binding by ABCE1), but this interaction is lost because ADPNP is added and replaces ADP, forcing eEF3 into a fully closed state that does not support binding of PRE3 to POST1 ribosomes. The authors speculate that the nucleotides they observe in the nucleotide binding pockets of eEF3 are natively bound ATP, not ADPNP, but there is no clear proof for this. I fully understand that the cryo-EM analysis cannot be easily repeated without ADPNP, but the authors should discuss these concerns. Furthermore, eEF3-binding experiments were suggested in the presence of different antibiotics to support the authors' model of eEF3 interaction with different ribosomal states. For example, immunoprecipitation in the presence of CHX or TIG should yield different amounts of co-

immunoprecipitated ribosomes, which hopefully supports the model in Figure 7.

REFEREE REPORTS**Referee #3:****eEF3 Revision**

The authors have addressed all critical points except for two major aspects: 1) Biochemical and biophysical in vitro studies; 2) Structural data - page 17-18 and Fig. 7. Overall, the revised manuscript has been largely improved, and addressing these two pending issues would strengthen its conclusions.

Major concern 1 - biochemical and biophysical in vitro studies:

The authors do not state that (or how much) ATP was used in the in vitro experiments. I assume that they simply forgot to mention it. This point is critical, because the ATP-occluded has been described as ribosome-bound in this and previous studies.

The authors replied: All experiments where eEF3 is present were carried out in the presence of ATP; we apologize for the unintended omission. This information has been now included in Materials and Methods, p. 39.

Remaining concern: However, the ATP concentrations used in the kinetic experiments were not physiological and unexpectedly low, i.e. the authors used a 10-fold molar excess of ATP (20 or 40 μM) relative to eEF3 (2 or 4 μM). In the presence of ribosomes, eEF3 hydrolyses approximately 900-1000 ATP/min (kcat; Sarthy et al., 1997; Uritani and Miyazaki, 1988). If we now assume that only 0.35 μM (concentration of ribosomal complexes) of eEF3 exhibit the ribosome-activated ATPase activity, the total ATP would be hydrolyzed after 4 to 8 sec. This may not be critical for experiments with very fast kinetics (Fig 1B, C) but everything beyond this time span runs on GTP (Fig. 1D, 2C, E, especially 2E, EV1). Given the ATP vs. GTP concentrations in the cell, this is not physiological.

In their reply, the authors mention a 'rather high' GTP concentration, but how high is the concentration exactly? The authors should discuss these important experimental details in the main text. Mentioning the ATP concentration in Material & Methods only is, in my opinion, not sufficient. I encourage the authors to repeat the above-mentioned critical experiments in the presence of a higher ATP concentration. Alternatively, they should find convincing evidence to legitimate a 'GTP-fueled' kinetic analysis of a translational ATPase and mention it in the main text.

We optimized the ATP concentration in our experiments and 20-40 μM ATP is the optimum concentration for both translation and eEF3 function. We observed that at higher ATP concentrations the activity of eEF1 was somewhat reduced for unknown reasons and hence avoided using too high concentrations that did not improve eEF3 activity. There is no ATP depletion during the experiment, because we use ATP/GTP regeneration system in all experiments (phosphoenol pyruvate and pyruvate kinase; their concentrations, and also that of GTP, 0.5 mM, are given on p. 38). The concentration of the regeneration system is optimized to convert ADP/GDP to ATP/GTP in the milliseconds range, thus on the time scale of our experiments there is always ATP in the reaction. From the data of Uritani and Miyazaki (1987), the kcat/Km of eEF3 for ATP and GTP differ by a factor of 2.6, suggesting that the nucleotide specificity of eEF3 is not very high and thus eEF3 is not a pure translational ATPase, but rather an enzyme that can be fueled by both ATP and GTP.

Major concern 2 - structural data, page 17-18 and Fig. 7:

The authors should discuss that the use of AMPPNP and a chemical crosslinker might stabilize non-physiological states of the 80S-eEF3 complex or even complexes that are never formed in vivo. This is reminiscent of the situation with ABCE1 in translation initiation complexes, which in structural studies is always bound to 40S subunits in the presence of AMP-PNP (see e.g. Simonetti et al., 2020 or Heuer et al., 2017), but dissociates upon ATP addition (see e.g. Simonetti et al., 2020). A new BioRxiv preprint (Kratzat et al., 2020) now shows ABCE1 in initiation complexes in a new state, possibly manifesting its role in translation initiation, but certainly showing that AMP-PNP-containing samples cannot be used to extend structural data to functional knowledge without validation by biochemical experiments. If this cannot be ruled out by convincing arguments or references, the authors should relativize the respective statements. Alternatively, cryo-EM

experiments (low resolution should be sufficient) could be performed using ex-vivo samples lacking AMP-PNP. To substantiate their statements, the authors could also consider biochemical (e.g. co-immunoprecipitation) or biophysical (e.g. fluorescence-based) eEF3-binding studies on defined 80S complexes (in the presence of antibiotics).

The authors replied: There appears to be a misunderstanding as to how the complexes were prepared. The cells were not lysed in the presence of AMP-PNP, nor were they applied to the beads or washed in the presence of AMP-PNP ...only the elution (by protease) of the complexes was performed in a buffer containing AMP-PNP (as stated in the methods on page 40). Since the lysing and application procedure takes over 3 hours, we think that it is unlikely that complexes will be present that were never formed in vivo - more likely that complexes will have disassembled during this time that we don't observe (perhaps explaining why we do not observe for example the ternary complex delivery state; see the penultimate comment of reviewer #2 above). Moreover, we have also performed the elution without the presence of AMP-PNP and observed similar results (now included as new Appendix Figure S2F), which suggests that the purification of ribosomal complexes does not require AMP-PNP. Collectively, we suspect that the molecule we see in the active site is ATP and not AMP-PNP however, we do not have the resolution to distinguish between these two molecules and are thus careful not to favor one over the other in the text, for example, as on page 12 "We note that electron density consistent with ATP (or ADPNP) was observed in the active sites of the ABC1 and ABC2 nucleotide".

Remaining concern: The authors included an SDS-PAGE/Coomassie showing that protease elution of eEF3 from the beads during immunoprecipitation in the absence and presence of ADPNP yields the same proteins. Indeed, this excludes the elution of proteins that were not bound to eEF3 in the absence of ADPNP. But why was ADPNP then added during elution? This is somewhat surprising as the authors could have performed the cryo-EM analysis on a native sample but decided to add an artificial nucleotide.

We did of course perform the analysis initially on the native sample without ADPNP, however we observed no density for eEF3, only empty ribosomes. We considered different strategies as to how eEF3 could be stabilized on the ribosome, and decided to employ the non-hydrolysable ATP analog ADPNP. ADPNP was not added from the beginning of the purification process because the sample was isolated from a 15 L culture, which would require extremely large (and unrealistic) amounts of compound. We considered that addition during the three-hour TAP-Tag cleavage step might nevertheless help to prevent dissociation of eEF3 from the ribosome, at least during this step. At this stage, we had not considered the possibility to obtain so many different states with eEF3 bound, but rather aimed for a single native eEF3-80S complex. By chance, at the same time, we also managed to employ mild crosslinking conditions to the same sample without inducing aggregation. After a small data collection and processing we were excited to finally observe density for eEF3, and therefore these grids were immediately used for an extensive data collection aiming for a high-resolution structure. It is important to point out that the samples cannot be stored and must be immediately applied to cryo-grids otherwise the ligands dissociate in the absence of crosslinker, or aggregate in the presence of crosslinker. Thus, while retrospectively, we agree with reviewer #3 that one could have gone back to try to optimize the sample in the absence of ADPNP, we decided rather to move forward given that the only previous structure of eEF3

on the ribosome was in vitro reconstituted and low resolution, and therefore provided little insight into which state(s) eEF3 interacts with in the cell. While this has limited our insights into the role of ATP in the function of eEF3, we believe that such a study could be undertaken in the future, perhaps also by employing EQ mutants in the respective NBDs of eEF3 and analyzing these mutants structurally as well as biochemically.

Further, the addition of ADPNP during elution might have altered the conformation of eEF3 and its affinity for certain ribosomal states. For example, it may well be that eEF3 stably binds to the states PRE3 to POST1 (Figure 7) in a hybrid ATP/ADP state (similar to 80S binding by ABCE1), but this interaction is lost because ADPNP is added and replaces ADP, forcing eEF3 into a fully closed state that does not support binding of PRE3 to POST1 ribosomes. The authors speculate that the nucleotides they observe in the nucleotide binding pockets of eEF3 are natively bound ATP, not ADPNP, but there is no clear proof for this. I fully understand that the cryo-EM analysis cannot be easily repeated without ADPNP, but the authors should discuss these concerns.

Yes, the reviewer raises a good point that the presence of ADPNP in the elution buffer could lead to changes in hybrid ATP/ADP states into ATP/ADPNP states. In fact, we believe that this may in fact be the case, since such an occurrence would explain the discrepancy between our closed state in POST-2 and the finding that the presence of ADPNP prevents E-tRNA release by EF3. We thank the reviewer for this comment and have now included some discussion on this point on page 19.

Furthermore, eEF3-binding experiments were suggested in the presence of different antibiotics to support the authors' model of eEF3 interaction with different ribosomal states. For example, immunoprecipitation in the presence of CHX or TIG should yield different amounts of co-immunoprecipitated ribosomes, which hopefully supports the model in Figure 7.

Unfortunately, our pullouts are not quantitative and since we observe density for eEF3 in both rotated and non-rotated states, we would not want to pursue such experiments that rely on only differences in the quantity of ribosomes eluted. Better would be to generate defined conformational states in vitro and monitor eEF3 binding using some accurate fluorescent kinetic approach. However, such an assay is not trivial to establish and therefore we feel it is beyond the scope of this paper.

Thank you again for submitting the final revised version of your manuscript. I am pleased to inform you that we have now accepted it for publication in The EMBO Journal.

Corresponding Author Name: Daniel Wilson, Rachel Green, Marina V. Rodnina, Namit Ranjan

Journal Submitted to: The EMBO Journal

Manuscript Number: EMBOJ-2020-106449R1